# TIPS: A Text-Image Pairs Synthesis Framework for Robust Text-based Person Retrieval

## Abstract

Text-based Person Retrieval (TPR) faces critical challenges in practical applications, including zero-shot adaptation, few-shot adaptation, and robustness issues. To address these challenges, we propose a Text-Image Pairs Synthesis (TIPS) framework, which is capable of generating high-fidelity and diverse pedestrian text-image pairs in various real-world scenarios. Firstly, two efficient diffusion-model fine-tuning strategies are proposed to develop a Seed Person Image Generator (SPG) and an Identity Preservation Generator (IDPG), thus generating person image sets that preserve the same identity. Secondly, a general TIPS approach utilizing LLM-driven text prompt synthesis is constructed to produce person images in conjunction with SPG and IDPG. Meanwhile, a Multi-modal Large Language Model (MLLM) is employed to filter images to ensure data quality and generate diverse captions. Furthermore, a Test-Time Augmentation (TTA) strategy is introduced, which combines textual and visual features via dual-encoder inference to consistently improve performance without architectural modifications. Extensive experiments conducted on TPR datasets demonstrate consistent performance improvements of three representative TPR methods across zero-shot, few-shot, and generalization settings.

## 1 Introduction

Text-based Person Retrieval (TPR) Li et al. (2017) aims to precisely locate individuals in image galleries using natural language descriptions and addresses identity recognition challenges in vision-limited scenarios through cross-modal alignment. Although feature learning frameworks Jiang & Ye (2023); Qin et al. (2024); Bai et al. (2023) have advanced and improved retrieval accuracy on benchmark datasets Li et al. (2017); Ding et al. (2021); Zhu et al. (2021), two critical challenges remain unresolved: rapid adaptation to new domains and enhancing robustness in practical applications.

As shown in Figure 1a, some existing methods Yang et al. (2023); Shao et al. (2023); Tan et al. (2024) have attempted data-level solutions, but fundamental limitations persist. Unlike methods relying on labor-intensive manually labeled datasets, these methods focus on automatically synthesizing large-scale datasets to enhance retrieval adaptability in novel scenarios. However, these approaches usually incorporate collected real person images into the final datasets, limiting their extensibility and scenario diversity. Meanwhile, methods that combine real texts with generative models Goodfellow et al. (2020); Rombach et al. (2022) often yield low-quality outputs that are inconsistent with target distributions. Recent studies based on Stable Diffusion Rombach et al. (2022) for dataset construction, such as MALS Yang et al. (2023), suffer from poor image quality and text-image alignment. Although newer models like Flux Labs (2024) enhance generative fidelity, their emphasis on high-definition and aesthetic outputs still fails to align with the multi-resolution distributions commonly observed in real-world scenarios (see Figure 1b). Additionally, these methods do not consider scenarios with limited labeled data in the target domain, thus resulting in fixed and independent data expansion processes.

To address these challenges, we first focus on the visual style of generated images and propose a parameter-efficient diffusion-model fine-tuning approach for generating clarity-controllable person images. Traditional few-shot multi-resolution fine-tuning often fails to achieve multi-scale generative capabilities. In comparison, our innovation lies in conditioning on image width and height parameters during fixed-resolution training, enabling dynamic control of image clarity at a fixed physical resolution during inference. Accordingly, we develop a Seed Person Image Generator (SPG),

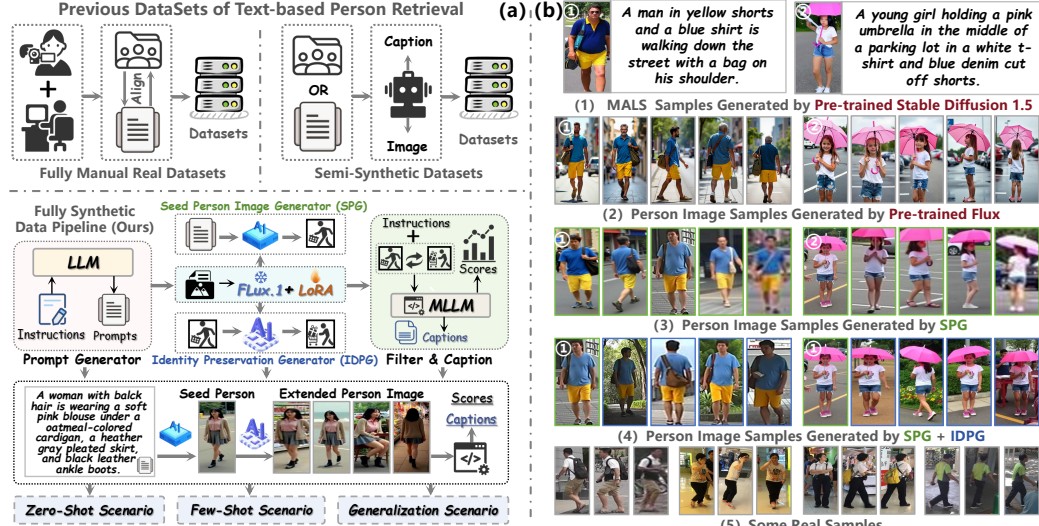

Figure 1: Ten data groups with an MLLM identity-consistency score of 8. For each group, the left image shows the seed person image generated by SPG, and the right image shows the extended person image generated by IDPG.

as shown in Figure 1b, which accurately adjusts blur levels while maintaining batch-generation capability. Furthermore, when limited target-domain annotations are available, SPG can utilize them to generate person images that are better aligned with the target-domain distribution. However, person images generated multiple times using the same prompt, similar to the pretrained Flux, may exhibit inconsistencies in appearance and identity. To preserve person identity, we design an Identity Preservation Generator (IDPG), which leverages efficient LoRA-based fine-tuning to enhance the contextual identity-preserving capability. Consequently, IDPG expands multiple images of the same identity by taking reference images and textual variations as inputs. By combining SPG and IDPG, we achieve for the first time in the field the ability to generate identity-consistent image sets solely from textual descriptions. As shown in Figure 1b, this method significantly surpasses existing methods, and can be comparable to real images in terms of fidelity and diversity.

Secondly, we construct a Text-Image Pairs Synthesis (TIPS) framework that integrates Large Language Model (LLM) Yang et al. (2025) with SPG and IDPG to automatically synthesize diverse person images. Moreover, a Multimodal Large Language Model (MLLM) Bai et al. (2025) is further employed to score generated images across multiple dimensions, ensuring high-quality output. Subsequently, MLLM will generate captions for the filtered images and create the image-text pairs needed for training. Finally, we introduce a test-time augmentation (TTA) strategy to improve retrieval accuracy by fusing text queries and synthesized visual features through dual-encoder inference without altering the TPR model architecture.

Comprehensive experiments conducted on CUHK-PEDES (CUHK) Li et al. (2017), ICFG-PEDES (ICFG) Ding et al. (2021), and RSTPReid (RSTP) Zhu et al. (2021) datasets across zero-shot, few-shot, and generalization settings consistently demonstrate performance improvements for three representative methods Jiang & Ye (2023); Qin et al. (2024); Bai et al. (2023). In low-data scenarios (as low as 1% labeled data), the TIPS framework achieves over 85% average performance gains. Moreover, in zero-shot scenarios, it also maintains significant advantages compared to large-scale synthetic datasets based on real images. Our main contributions are summarized as follows:

- Two generators, namely SPG and IDPG, are proposed based on novel parameter-efficient fine-tuning methods, achieving the first text-only generation of identity-consistent image sets in the field of TPR.

- By integrating LLM, MLLM, SPG, and IDPG, a novel TIPS framework is constructed to automate the generation of fully synthetic TPR datasets, where high-fidelity and diverse person images are aligned with real-world scenarios.

- A universally applicable TTA strategy is introduced to enhance retrieval accuracy without requiring structural modifications.
- Extensive experiments on three benchmark datasets demonstrate consistent and comprehensive performance improvements.

## 2 RELATED WORK

### 2.1 TEXT-BASED PERSON RETRIEVAL

Recent advances leverage vision-language pretrained (VLP) models Radford et al. (2021); Li et al. (2021; 2022) through two main strategies: cross-modal attention interaction Bai et al. (2023); Yang et al. (2023); Ergasti et al. (2024) and cross-modal-free approaches Jiang & Ye (2023); Liu et al. (2025). Interaction methods improve modality alignment by computing feature attention scores during inference, but increase computational complexity. Additionally, some studies have aimed at improving TPR methods through synthetic data. Specifically, LUPerson-T Shao et al. (2023) and LUPerson-M Tan et al. (2024) focus on generating synthetic textual descriptions based on the large-scale person dataset LUPerson Fu et al. (2021) to construct pre-training datasets. In contrast, MALS Yang et al. (2023) attempts to directly generate person images and texts for pre-training dataset construction. However, it still requires original annotation texts to guide the diffusion models, and more crucially, the generated person images are of low quality and identity information is lost. All these methods rely on incorporating original real data into the final pre-training datasets, leading to privacy-sensitive and insufficient diversity issues. Moreover, they focus primarily on pre-training scenarios, instead, we propose a fully synthetic TPR data synthesis paradigm aiming to enhance the practical utility of TPR methods in various scenarios.

### 2.2 DIFFUSION MODELS

Diffusion models Ho et al. (2020) have become the dominant framework for image generation, excelling in tasks such as text-to-image synthesis Saharia et al. (2022b); Podell et al. (2023); Ramesh et al. (2022), image-to-image translation Saharia et al. (2022a); Huang et al. (2025); Xie et al. (2023), and controllable generation Zhang et al. (2023); Ye et al. (2023); Qin et al. (2023). The introduction of Latent Diffusion Models (LDM) Rombach et al. (2022) has significantly improved text-image alignment and reduced computational costs through latent space operations. This advancement enables parameter-efficient fine-tuning methods, such as Low-Rank Adaptation (LoRA) Hu et al. (2022) and Adapter Houlsby et al. (2019), to be applied effectively for domain adaptation while preserving generation quality. Recently, combining diffusion models with transformer (DiT) Peebles & Xie (2023a) architectures has further enhanced scalability, leading to advanced models such as Stable Diffusion 3 Esser et al. (2024), PixArt Chen et al. (2024), and Flux Labs (2024). These models utilize flow matching Lipman et al. (2022) objectives to achieve state-of-the-art (SOTA) generation quality and exhibit strong multi-subgraph Hui et al. (2025) and contextual generation capabilities Tan et al. (2025). Inspired by these advancements, we first propose the SPG for high-fidelity person image generation, which efficiently embeds the LoRA into Flux, and then the IDPG is constructed to achieve identity preservation.

## 3 METHOD

As shown in Figure 2, the fully automated synthesis of TPR data consists of three interrelated components: LLM-driven text generation, person image generation, and image quality filtering with caption generation. We first describe the design and training of the person image generators SPG and IDPG in Section 3.1. Section 3.2 introduces the TIPS data synthesis framework. Additionally, an optional TTA module (see Section 3.3) is introduced during inference to integrate textual and visual cues to enhance the retrieval accuracy.

### 3.1 SPG AND IDPG

**Parametric Resolution Control via SPG.** Low-resolution data is prevalent in TPR, yet traditional fine-tuning methods struggle to generate domain-adaptive person images. Standard bucket-based

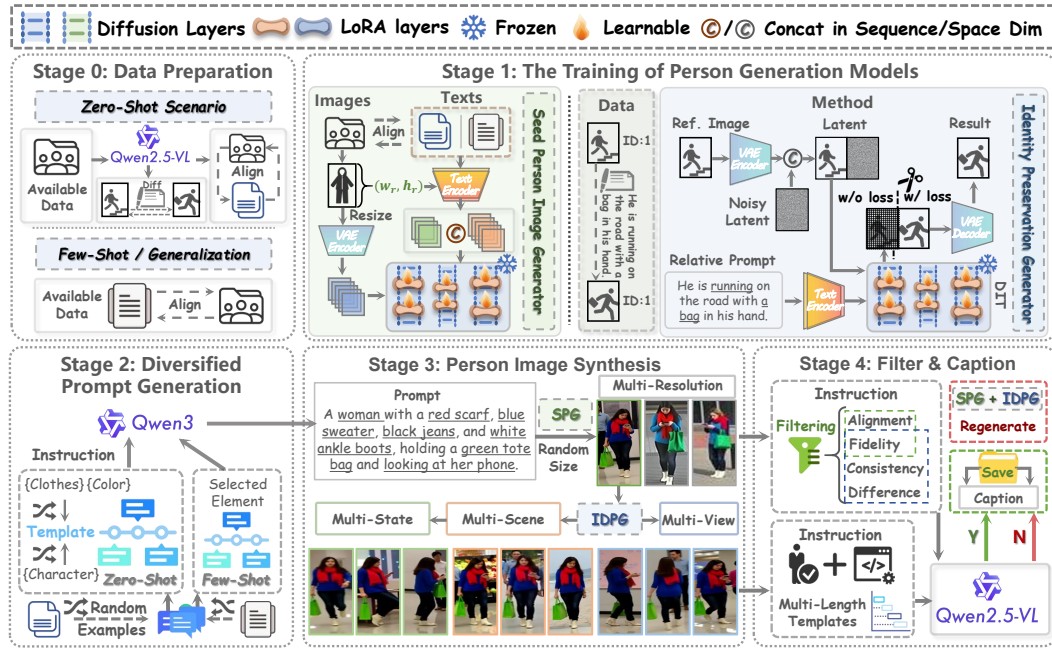

Figure 2: Overview of the TIPS framework pipeline with five stages. Stage 0: preparing data for different scenarios to train person-image generators, including image-text pairs for SPG and triplets for IDPG. **Stage 1**: separately training SPG and IDPG. **Stage 2**: generating diversified seed-person prompts via LLM. **Stage 3**: synthesizing seed-person images using SPG, then expanding these images using IDPG to create identity-consistent image sets. **Stage 4**: obtaining high-quality data through MLLM-based image filtering and caption generation.

training Peebles & Xie (2023b); Labs (2024) faces two limitations: firstly, it requires numerous samples for each resolution bucket (impractical for few-shot TPR), and secondly, it is constrained by the 32-pixel grid alignment in mainstream DiT-based models Peebles & Xie (2023a) such as standard Flux. To overcome these issues, our proposed SPG employs a parametric resolution control strategy. It is trained at a fixed physical resolution, conditioned on resolution parameters $(w_r, h_r)$ via the text encoder, enabling continuous resolution control during inference without architectural constraints.

**Identity Preservation via IDPG.** Although SPG can generate person images that are consistent with the target-domain distribution, due to the stochastic nature of diffusion models, different initial noises may produce images with differing identities even when the same prompt is used. To achieve identity-preserving person image generation, inspired by dual-image generative models Hui et al. (2025); Tan et al. (2025), we design the IDPG, which concatenates reference person image features with noisy target person images at the latent level, effectively introducing image conditions without modifying the model structure. Guided by difference prompts, the model can predict the target person images, thus can achieve text-based expansion of images with consistent identities once a reference image is given.

To adapt to different application scenarios, we construct different training data. Specifically, without any labeled data in the target domain, based on a small set of collected real-person images, we first utilize an MLLM to generate captions for training the SPG, and also generate differential captions for pairs of images of the same identity to train the IDPG. Secondly, when labeled data from the target domain is available, the existing image-text pairs can directly train the SPG to generate images more aligned with the target domain distribution.

**Preliminaries.** Both SPG and IDPG are constructed upon the Flux architecture, a Diffusion Transformer (DiT) model. In the standard formulation, the model takes a noisy target latent $Z_{\text{tgt}}$ (derived from the input image via VAE) and a condition embedding as inputs. The core mechanism involves cross-attention layers where the latent features interact with the conditional embeddings. Specifi-

cally, the Query ($Q$), Key ($K$), and Value ($V$) are projected as follows:

$$Q = \mathbf{W}_Q \cdot \phi(Z), \quad K = \mathbf{W}_K \cdot \tau(P), \quad V = \mathbf{W}_V \cdot \tau(P) \tag{1}$$

where $\mathbf{W}_{\{Q,K,V\}}$ are learnable projection matrices, $\phi(\cdot)$ denotes the patchification of image latents, and $\tau(\cdot)$ represents the pre-trained text encoder. $P$ denotes the input prompt tokens. To enhance robustness and controllability without sacrificing pre-trained capabilities, we modify the inputs for SPG and IDPG as detailed below.

**Seed Person Image Generator.** Standard fine-tuning at fixed resolutions limits the diversity of generated aspect ratios. To address this, SPG conditions the generation on both textual descriptions and explicit resolution parameters $(w_r, h_r)$. Crucially, rather than initializing a new encoder, we reuse the pre-trained text encoder $\tau(\cdot)$ to process these resolution constraints. We formulate the resolution parameters as textual tokens $P_{\text{size}}$ (e.g., strings representing width and height) and concatenate them with the content prompt $P_{\text{txt}}$. As shown in Eq. 2, the composite condition $C_{\text{spg}}$ is derived as:

$$C_{\text{spg}} = [\tau(P_{\text{size}}) \,;\, \tau(P_{\text{txt}})] \tag{2}$$

where $[\cdot \,;\, \cdot]$ denotes concatenation along the sequence dimension. By feeding $C_{\text{spg}}$ into the Key and Value projections in Eq. 1, SPG effectively leverages the model's pre-trained semantic space to dynamically control the clarity and aspect ratio of the generated person images ($Z_{\text{tgt}}$), ensuring high-fidelity outputs even when the model acts on fixed-size tensors.

**Identity Preservation Generator.** To ensure identity consistency across generated samples, IDPG requires a reference signal in addition to the text prompt. We adopt a dual-latent strategy that modifies the query input $Z$.

Given a reference person image $I_{\text{ref}}$, we encode it into a reference latent $Z_{\text{ref}}$ using the fixed VAE. Instead of altering the model architecture, we explicitly expand the visual context by concatenating the reference latent with the noisy target latent in the spatial dimension:

$$Z_{\text{idpg}} = [Z_{\text{ref}} \,;\, Z_{\text{tgt}}] \tag{3}$$

Consequently, the input to the attention mechanism in Eq. 1 becomes $Z = Z_{\text{idpg}}$. This allows the self-attention layers to directly attend to fine-grained identity features in $Z_{\text{ref}}$ while denoising $Z_{\text{tgt}}$.

**LoRA Fine-tuning.** To efficiently adapt to these distributions, we employ Low-Rank Adaptation (LoRA) Hu et al. (2022). We freeze the pre-trained weights $\mathbf{W}$ and inject trainable low-rank matrices $B$ and $A$. The forward pass is updated as $\mathbf{W}'x = (\mathbf{W} + \beta\gamma BA)x$, where $\beta$ is a hyperparameter controlling the adaptation strength and $\gamma$ is a learnable layer-specific scaling factor. For IDPG, the flow-matching objective Lipman et al. (2022) is calculated exclusively on the target person region ($Z_{\text{tgt}}$), ensuring the reference features act purely as stable conditions.

After training, SPG generates person images of desired distributions and adjustable clarity by combining text prompts with specified resolution conditions $(w_r, h_r)$. Then, these seed images from the SPG are fed into the IDPG together with differential prompts to generate additional images that preserve the same identity.

## 3.2 THE TIPS FRAMEWORK

As previously discussed, high-quality TPR datasets are both scarce and essential, while existing synthetic methods fail to adequately meet practical requirements. To overcome these limitations, we propose an automated TIPS framework (illustrated in Figure 2; See Appendix A for details), structured into three stages: S1) diversified prompt generation driven by LLM, S2) high-quality person image synthesis, and S3) image quality filtering and caption generation.

In S1, we focus on ensuring textual diversity and domain relevance. When target-domain data is absent, input instructions for the LLM randomly combine descriptive elements from predefined candidate lists and supplement these with three randomly selected examples from SPG's training captions, ensuring both output stability and maximized diversity. In scenarios with limited target-domain data, we enhance domain consistency by extracting three sentences from available texts and recombining selected elements into stylistically coherent new sentences.

In S2, we leverage the trained SPG and IDPG to synthesize person images based on generated prompts. Initially, SPG creates a seed person image, which, upon passing quality criteria, serves as

input to IDPG. IDPG then expands the seed image by generating additional images that maintain the identity across varied perspectives, contexts, and states. This two-step generation process addresses the critical issue of identity consistency and enables extensive diversity. It is important to note that the diversity of the synthesized dataset is driven by the rich semantic variations in the generated prompts, rather than the small set of real images. Since LoRA fine-tuning preserves the pre-trained base model's generalization capabilities, the generators can translate diverse textual descriptions into varied visual content while maintaining the learned surveillance style.

The final stage (S3) addresses the necessity of maintaining high data quality and textual alignment. All generated images undergo rigorous quality evaluation via an MLLM. Specifically, seed images are assessed on their prompt-image alignment and overall naturalness and fidelity. Images not meeting standards are regenerated until they pass. IDPG-generated images are further evaluated for identity and outfit consistency with the seed image, as well as required attribute variation, ensuring high-quality, identity-consistent image sets. Subsequently, retained images receive diverse textual descriptions through the MLLM, utilizing a varied set of long and short sentence templates to enrich caption style and ensure output stability.

### 3.3 TEST-TIME AUGMENTATION

To comprehensively achieve the goal of robust TPR, it is essential to address challenges not only through data-driven enhancements at the training stage but also via optimization at the inference stage. Complementing the TIPS framework, we introduce a general-purpose TTA strategy that leverages the generative priors of SPG to bridge the cross-modal gap during testing. Specifically, the SPG enables TTA by synthesizing candidate images from text queries. Conventional TPR methods employ dual encoders trained with identity-aware contrastive loss Zhang & Lu (2018); Jiang & Ye (2023) to optimize cross-modal alignment and intra-modal consistency. As shown in Figure 3a, standard TPR inference processes query text $t_q$ and gallery images $\mathbf{I} = (i_1, \ldots, i_N)$ through dual encoders to extract text features $f_t$ and image features $\mathbf{F}_i = (f_{i1}, \ldots, f_{iN})$. Global representations $f_t^g$ (text) and $\mathbf{F_i^g} = (f_{i1}^g, \ldots, f_{iN}^g)$ (images) compute similarity scores:

$$S_j = \frac{f_t^g \cdot f_{ij}^g}{\|f_t^g\| \, \|f_{ij}^g\|}, \quad j = 1, \ldots, N. \tag{4}$$

To refine initial rankings, some methods Bai et al. (2023); Yang et al. (2023) apply transformer-based reranking to obtain top-K candidates. However, in these methods, the intra-modal consistency cannot be fully utilized during inference. To alleviate this issue, we design a feature fusion method with a TTA strategy. As shown in Figure 3b, our TTA extension includes three phases: 1) Generate preview image $i_p$ from $t_q$ using SPG; 2) Extract $i_p$'s visual feature $f_p^g$ and compute hybrid query:

$$f_q^g = \alpha f_t^g + (1 - \alpha) f_p^g, \tag{5}$$

where $\alpha$ is a hyperparameter controlling the synthesized image's retrieval weight, with larger values reducing the contribution of $i_p$; 3) Recompute similarities using $f_q^g$, and optionally rerank to get updated top-K candidates. Therefore, our method enhances the intra-modal consistency

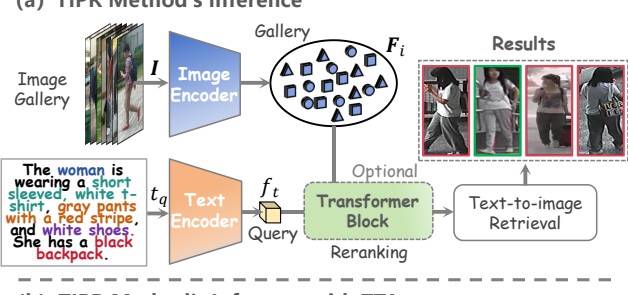

Figure 3: Inference pipelines (a) without and (b) with TTA.

by leveraging the dual encoders' latent alignment from contrastive training without architectural changes. Through empirical calibration ($\alpha \in [0, 1]$), we balance cross-modal matching and visual consistency for attaining robust performance.

# 4 EXPERIMENTS

## 4.1 EXPERIMENTAL SETTINGS

**Implementation Details.** We evaluate three practical settings for TPR: (i) zero-shot—no paired annotations in the target domain; (ii) few-shot—only a small set of labeled samples; and (iii) cross-domain generalization—training annotations come from a different domain.

For the zero-shot setting, where models are trained exclusively on synthesized data and directly evaluated on target benchmarks without accessing any domain-specific training samples, we uniformly select 100 IDs from five datasets (CUHK03 Li et al. (2014), CUHK02 Li & Wang (2013), Market-1501 Zheng et al. (2015), MSMT17 Wei et al. (2018), and VIPER Gray & Tao (2008)), ensuring that these IDs do not overlap with the image sources in the TPR artificial dataset. For each selected ID, three images are randomly chosen, totaling 300 images. Each image is captioned by the MLLM with two distinct captions, which are used to train the SPG. Additionally, image pairs from the same ID (but different images) are used to train the IDPG, utilizing their differential captions. In cases where annotated data is available, SPG training leverages the provided image-text pairs.

All experiments are performed on two H800 GPUs, using Qwen3-32B Yang et al. (2025) as the LLM and Qwen2.5VL-32B Bai et al. (2025) as the MLLM. Both the SPG and IDPG are based on FLUX.1-dev Labs (2024), with a LoRA rank set to $r=32$. The input images are resized to $192 \times 384$. During SPG training, 20% of the samples have width–height conditions randomly dropped to enhance robustness. Similarly, for IDPG training, 10% of the differential captions are randomly omitted to improve the model's ability to generate identity-preserving images with diverse representations. Each experiment generates 40k independent seed prompts, each paired with resolution parameters learned by the SPG. The SPG generates one image of size $192 \times 384$ per seed prompt through 28 sampling steps. These images are then processed by the IDPG, which generates four additional images of the same identity using random difference prompts. All generated images undergo a filtering step by the MLLM, which checks for consistency between the seed and expanded images through binary classification. Other dimensions are scored on a 1–10 scale, with a minimum required score of 9. Images that do not meet these criteria are regenerated. Subsequently, the filtered images are paired with two captions generated by MLLM using randomly selected templates. In total, this process yields 40k IDs, 200k images, and 400k image-text pairs.

For evaluation, three representative TPR methods (IRRA Jiang & Ye (2023), RDE Qin et al. (2024), Rasa Bai et al. (2023)) are evaluated on CUHK, ICFG, and RSTP datasets, following their original configurations. Training in the few-shot scenario comprises two phases: 1) synthetic data training with original hyperparameters, 2) fine-tuning with real data, with epochs and learning rates halved. In TTA, the value of $\alpha$ is set to 0.6. (Further details are provided in Appendix C.3.)

**Evaluation Metrics.** Retrieval performance is measured by Rank-k (R@k) accuracy and mean average precision (mAP), where R@k indicates the proportion of queries with correct matches in the top-k results, and mAP averages precision over all queries. Fréchet Inception Distance (FID) Heusel et al. (2017) assesses the distributional similarity between training and testing images.

Table 1: Zero-shot retrieval performance of different methods under various pre-training data configurations.

| Method | Pre-training Data | Scale | CUHK | | ICFG | | RSTP | |
|---|---|---|---|---|---|---|---|---|
| | | | R@1 | mAP | R@1 | mAP | R@1 | mAP |
| CLIP | – | – | 12.65 | 11.15 | 6.67 | 2.51 | 13.45 | 10.31 |
| IRRA | MALS | 1.5M | 19.21 | 18.72 | 7.88 | 3.49 | 22.50 | 16.94 |
| | LUperson-T | 400K | 20.06 | 19.24 | 10.46 | 4.11 | 22.10 | 16.79 |
| | | 1M | 22.03 | 21.64 | 12.31 | 4.98 | 22.95 | 17.23 |
| | LUperson-M | 400K | 48.07 | 44.12 | 27.35 | 13.95 | 42.95 | 32.46 |
| | | 4M | **53.23** | 47.66 | **33.27** | **17.33** | 48.50 | 38.96 |
| | **Ours** | 400K | 52.89 | **47.81** | 33.16 | 17.15 | **48.65** | **39.04** |
| RaSa | MALS | 1.5M | 21.66 | 21.02 | 9.72 | 3.95 | 26.20 | 20.39 |
| | LUperson-T | 400K | 22.41 | 21.49 | 12.22 | 5.80 | 25.65 | 20.17 |
| | | 1M | 24.33 | 23.79 | 14.04 | 6.59 | 26.40 | 20.46 |
| | LUperson-M | 400K | 50.72 | 46.67 | 29.34 | 15.86 | 46.95 | 36.29 |
| | | 4M | **55.73** | **50.06** | **35.15** | 19.13 | 52.25 | 41.56 |
| | **Ours** | 400K | 55.44 | 49.89 | 35.07 | **19.27** | **52.50** | **41.61** |

## 4.2 QUANTITATIVE RESULTS

In this section, we comprehensively evaluate the proposed framework's capability in addressing realistic TPR applications through three simulated settings, and validate the effectiveness of the proposed TTA.

**Zero-shot Scenario.** In the zero-shot scenario, we select IRRA (without reranking) and RaSa (with reranking) as the baseline models. By default, no target-domain data is used in this scenario. Each model is trained using data expanded by the corresponding TIPS framework and then directly evaluated on all three test sets. The performance results are presented in Table 1, where both methods demonstrate consistent trends. Compared with models such as CLIP, which are not pretrained on pedestrian data, our pretrained models exhibit significant improvements. Addition-

Table 2: Few-shot retrieval performance of different methods under various pre-training data configurations.

| | Pre-training Data | CUHK | | ICFG | | RSTP | |
|---|---|---|---|---|---|---|---|
| | | R@1 | mAP | R@1 | mAP | R@1 | mAP |
| IRRA | – | 34.44 | 32.57 | 19.73 | 10.20 | 30.70 | 24.65 |
| | MALS | 38.61 | 35.66 | 22.18 | 12.27 | 34.70 | 27.16 |
| | LUperson-M | 53.84 | 47.92 | 39.78 | 21.24 | 50.90 | 39.50 |
| | **Ours** | **55.73** | **49.72** | **45.94** | **24.75** | **54.30** | **42.00** |
| RDE | – | 34.73 | 32.89 | 19.12 | 10.61 | 30.40 | 23.41 |
| | MALS | 39.38 | 36.27 | 23.56 | 12.53 | 35.80 | 29.90 |
| | LUperson-M | 55.70 | 49.67 | 41.96 | 22.69 | 52.40 | 39.91 |
| | **Ours** | **58.90** | **52.62** | **47.15** | **25.77** | **56.85** | **41.45** |
| RaSa | – | 45.92 | 38.54 | 21.16 | 5.21 | 38.85 | 24.27 |
| | MALS | 47.43 | 42.44 | 24.19 | 12.94 | 41.20 | 32.98 |
| | LUperson-M | 57.50 | 51.02 | 42.97 | 22.19 | 56.05 | 44.41 |
| | **Ours** | **60.95** | **53.89** | **49.11** | **26.67** | **61.25** | **48.24** |

ally, to illustrate the high quality of data generated by our TIPS framework, we also compare it against the synthetic pedestrian image dataset MALS Yang et al. (2023) and the real-image-based textual synthetic datasets LUperson-T Shao et al. (2023) and LUperson-M Tan et al. (2024). Due to the low quality of synthetic images and the lack of consideration for diverse resolutions and identity characteristics, MALS yields the poorest results. For the datasets utilizing real images, LUperson-T and LUperson-M, limited variation in images of the same identity sourced from the same video significantly hampers their performance relative to our method at the same scale. Even when training with the complete dataset, our method achieves comparable performance to the best-performing LUperson-M model, while using only 10% of the data volume.

**Few-shot Scenario.** We simulate scenarios with limited annotations by subsampling 1% of training IDs from the full datasets to validate the effectiveness of the TIPS framework under extremely limited samples. Each group of experiments utilizes identical subsampled data to ensure a fair comparison. Table 2 summarizes comparative results using three methods across three datasets, demonstrating that the expanded data using the TIPS framework achieves the best results in few-shot conditions. Compared with baseline models without pretraining, the Rank-1 performance of IRRA, RDE, and RaSa methods on the three datasets improves on average by 90.51%, 101.07%, and 74.16%, respectively, reaching practical usability. Compared to other full-scale pretrained datasets, the TIPS framework achieves the best performance using the least data. The results indicate that the proposed SPG can ensure that the generated TPR data align well with the current domain distribution,

Table 3: Retrieval performance using different source and target domain data before and after data expansion.

| Training Data | Source | Target | | | | | |
|---|---|---|---|---|---|---|---|
| | | CUHK | | ICFG | | RSTP | |
| | | R@1 | mAP | R@1 | mAP | R@1 | mAP |
| Raw | CUHK | 73.42 | 65.97 | 42.42 | 21.77 | 53.30 | 39.64 |
| | ICFG | 33.46 | 31.56 | 63.45 | 38.04 | 45.30 | 36.83 |
| | RSTP | 32.80 | 30.29 | 32.30 | 20.54 | 60.40 | 48.11 |
| **Ours** | CUHK | **75.80** | **68.55** | 47.80 | 25.60 | 57.85 | 42.89 |
| | ICFG | 47.17 | 42.73 | **65.98** | **40.16** | 53.70 | 41.08 |
| | RSTP | 45.83 | 42.09 | 45.51 | 27.74 | **64.90** | **50.32** |

*Note: "Raw" indicates training on source dataset only; "Ours" indicates training on source dataset + TIPS-expanded data.*

thus obtaining the optimal performance. This aspect holds substantial practical value, as annotating just 1% of the data (e.g., 31 IDs for ICFG, 37 IDs for RSTP) is easily achievable in real-world scenarios.

**Generalization Scenario.** Generally speaking, diversified training data can enhance model robustness by improving feature learning and the ability to handle out-of-distribution samples. Moreover, our framework indeed achieves controllable diversity through high-quality samples. For example, using IRRA as the TPR method, and training with complete source-domain data, cross-domain evaluation results (Table 3) demonstrate that our expanded data significantly improves the performance of all experiments. The results also indicate that even with full data, expanded training can significantly improve non-cross-domain performance, with an average Rank-1 increase of 3.14%. Improvements are even greater for cross-domain performance, with an average Rank-1 increase of 9.71%.

**Effectiveness of TTA.** Under the same few-shot settings, experiments are conducted on IRRA and RaSa to validate the effectiveness of the TTA strategy, and the results are shown in Table 4. The results demonstrate that TTA significantly improves performance across various settings without modifying the model structure. Note that TTA is an optional module, and combining data expansion with TTA can yield maximum improvements. Specifically, IRRA achieves Rank-1 improvements of 23.49%, 28.44%, and 25.70% on CUHK, ICFG, and RSTP, respectively, while RaSa improves by 17.09%, 29.68%, and 23.55%, respectively. TTA can be disabled if maximal inference efficiency is desired.

Table 4: Performance with and without data expansion (Data) and TTA. A ✓ indicates the component is enabled.

| | Setting | | CUHK | | ICFG | | RSTP | |
|---|---|---|---|---|---|---|---|---|
| | Data | TTA | R@1 | mAP | R@1 | mAP | R@1 | mAP |
| IRRA | | | 34.44 | 32.57 | 19.73 | 10.20 | 30.70 | 24.65 |
| | | ✓ | 41.78 | 38.01 | 27.75 | 13.17 | 36.55 | 28.51 |
| | ✓ | | 55.73 | 49.72 | 45.94 | 24.75 | 54.30 | 42.00 |
| | ✓ | ✓ | **57.93** | **51.54** | **48.17** | **26.06** | **56.40** | **43.61** |
| RaSa | | | 45.92 | 38.54 | 21.16 | 5.21 | 38.85 | 24.27 |
| | | ✓ | 50.13 | 45.44 | 28.64 | 13.42 | 44.30 | 35.83 |
| | ✓ | | 60.95 | 53.89 | 49.11 | 26.67 | 61.25 | 48.24 |
| | ✓ | ✓ | **63.01** | **55.64** | **50.84** | **28.15** | **62.40** | **49.72** |

### 4.3 QUALITATIVE RESULTS

Section 4.2 thoroughly discusses the substantial improvements achieved by the data expanded through the TIPS framework. In fact, to a large extent, these enhancements are attributed to the powerful person-image generation capabilities of the SPG and IDPG. Notably, by LoRA-based efficient tuning, our SPG and IDPG obtain zero-shot generation capabilities while effectively adapting to pedestrian image styles. As illustrated in Figure 4, by utilizing different prompts, the SPG generates comprehensive and highly realistic person images. In certain aspects such as diverse scenarios, varying weather conditions, and lighting situations, it even surpasses the level achievable in existing manually annotated datasets. When combined with the IDPG, the framework generates diverse images of the same identity. Thus, through appropriate LLM instruction design within the TIPS framework and leveraging MLLM's filtering and annotation mechanisms, these generators are capable of automatically producing high-quality data, consequently enhancing retrieval performance.

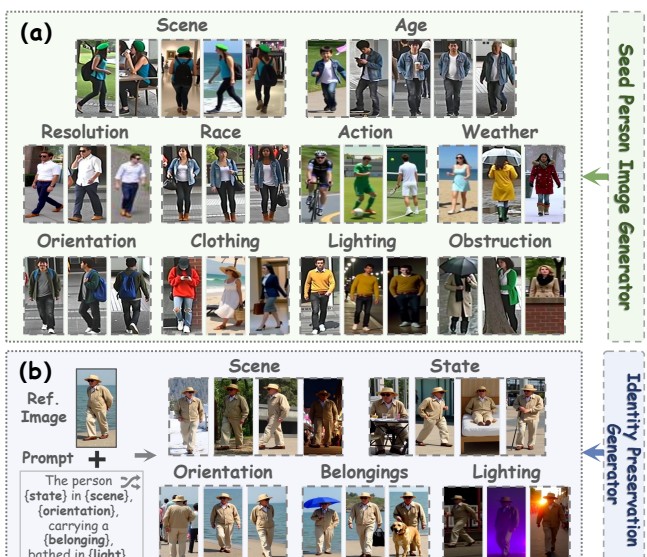

Figure 4: Zero-shot generation capability visualization.

## 4.4 ABLATION STUDIES

**Impact of Image Distribution.** Under the few-shot scenario, we employ IRRA on the CUHK dataset as the benchmark and compare six configurations of pre-training data to analyze how image distribution alignment influences retrieval performance: 1) no pre-training, 2) MALS dataset, 3) data generated by pretrained Flux, 4) LUPerson-M dataset, 5) data generated by SPG, and 6) data generated jointly by SPG and IDPG. Among these, configurations No.2, No.5, and No.6 contain an equal number of images; No.2 and No.5 cannot preserve identity, as each prompt generates only one image. The results in Table 5 reveal three key insights. Firstly, as long as the alignment between images and texts is ensured, any form of pre-training improves retrieval performance in low-data scenarios. Secondly, comparing configurations from No.2 to No.5 shows that, given high-quality generated pairs, better alignment with the target-domain distribution consistently leads to improved retrieval performance (See Appendix B for the theoretical analysis). Thanks to the SPG's ability to simulate target-domain data distribution with limited data, its performance even surpasses that of the larger-scale and real-image-based LUPerson-M dataset. Finally, comparing No.5 and No.6 demonstrates the effectiveness of incorporating identity-preserving generation in improving performance.

Table 5: Performance with different pre-training data.

| No. | Pre-training Data | FID ↓ | R@1 | R@5 | R@10 | mAP |
|-----|-------------------|-------|-------|-------|-------|-------|
| 1 | – | – | 34.44 | 58.35 | 68.41 | 32.57 |
| 2 | MALS | 105.69 | 38.61 | 61.62 | 72.30 | 35.66 |
| 3 | Pre-trained FLux | 116.74 | 42.40 | 65.51 | 75.18 | 39.04 |
| 4 | LUperson-M | 82.64 | 53.84 | 72.96 | 80.21 | 47.92 |
| 5 | SPG | **66.82** | 54.29 | 73.64 | 80.93 | 48.37 |
| 6 | **SPG+IDPG** | 68.07 | **55.73** | **75.04** | **82.84** | **49.72** |

## 5 CONCLUSION

In this paper, we propose the TIPS framework, a novel pipeline that automatically synthesizes high-quality text-image pairs to address core TPR challenges, including zero-shot and few-shot domain adaptation and robustness in practical scenarios. At its core are two efficient generators, SPG and IDPG, which create realistic, identity-consistent pedestrian images using minimal domain-specific data. Coupled with effective LLM-driven prompts and MLLM-based filtering, TIPS substantially enhances retrieval performance across various scenarios. Additionally, the proposed TTA method further improves retrieval accuracy without structural modifications. Extensive experiments on multiple benchmarks confirm TIPS's superiority, robustness, and practical applicability.

## ETHICS STATEMENT

The SPG and IDPG presented in this work have shown remarkable capabilities in generating high-fidelity, diverse, and identity-consistent images from textual descriptions. However, as with any generative model, there are inherent risks that must be carefully managed. Misuse of these technologies could lead to the creation of misleading or harmful content, infringing upon privacy, misrepresenting individuals, or enabling identity manipulation. To mitigate such risks, it is essential to adhere to ethical guidelines and exercise caution in their applications. All training data for the SPG and IDPG models have been sourced from publicly available datasets that have undergone rigorous checks to ensure they do not contain sensitive or private information. The datasets, including pedestrian image sets, have been carefully curated with fairness and privacy in mind. Furthermore, users of these models are encouraged to apply similar ethical considerations when using their own data to ensure that no sensitive or harmful content is generated. In light of potential misuse, we recommend the integration of digital watermarks in generated images, especially when models are made publicly available or open-sourced. Watermarking ensures traceability and accountability, helping to prevent the spread of deceptive images. If the generated data from this work is made open-source, digital watermarks will also be embedded to maintain the integrity and traceability of the content. Ultimately, we advocate for the responsible use of AI technologies, emphasizing the importance of transparency, privacy, and consent. By following ethical standards, we can contribute to the advancement of AI in a manner that promotes safety, trust, and societal well-being.

## REPRODUCIBILITY STATEMENT

In this study, to ensure the reproducibility of our approach, we provide the following key information from the main text and appendices:

1. **Algorithm.** We provide the architecture and core methods of TIPS in Figure 2 and Section 3. Additionally, we offer a more detailed practical implementation of TIPS in Appendix A and Figure A1, including the specific instructions used in all experiments. For further hyperparameter details, please refer to Appendix C.3.

2. **Source Code.** To enable complete reproduction of our work, we will release all relevant code as open-source after the review process is completed.

3. **Hyperparameters and Extended Analysis.** In Appendix D, we provide detailed experimental hyperparameters and conduct comprehensive ablation studies to validate the effectiveness of our proposed modules and strategies. Furthermore, Appendix E presents objective quantitative analyses and visual examples to substantiate the quality and diversity of the synthesized data.

4. **Theoretical Proofs.** We provide the core theoretical proofs supporting the effectiveness of the TIPS method in Appendix B.

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

## A    ADDITIONAL DETAILS OF DATA SYNTHESIS

### A.1    DETAILS OF DATA PREPARATION

To generate person images that are both realistic and aligned with expectations, we train the Seed Person Image Generator (SPG) and the Identity Preservation Generator (IDPG). Essential data preparation precedes this training.

Under the zero-shot setting, no usable person image–text pairs are available, so we construct such data entirely from scratch for SPG and build image–text–image triplets for IDPG. Specifically, we uniformly select 100 identities from five datasets (CUHK03 Li et al. (2014), CUHK02 Li & Wang (2013), Market-1501 Zheng et al. (2015), MSMT17 Wei et al. (2018), and VIPER Gray & Tao (2008)); these identities have no overlap with the manually annotated TPR datasets. Three images are randomly chosen for each identity, giving 300 images in total. An Multi-modal Large Language Model (MLLM) Bai et al. (2025) then applies the captioning instruction shown in Figure A1 to each image and produces two captions, thereby forming the image–text pairs used to train SPG.

In the few-shot or generalization setting, person image–text pairs are available and can be directly employed to train SPG. Because these pairs closely match the distribution of the test domain, the seed images generated by SPG in the few-shot scenario also move toward this distribution, which in turn improves the effectiveness of subsequent retrieval training.

For IDPG, identical training data are used across all settings, and each sample comprises a reference image, a relative description, and a target image. The image source remains the same 300 images collected from the five datasets. As each identity has three images, any two images of the same identity form three possible pairs. Each pair is passed to the MLLM, which follows the instruction below to generate the relative description.

> You will be provided with two images of the same person. Your task is to generate two **relative descriptions** that focus only on the differences in lighting, viewpoint, background scene, human status (pose, expression, etc.), and carried items. Do not describe any similarities and do not include explanations, reasoning, or preambles such as "Compared to the other image." Just output the differences. Use the following strict format:
> *[1] (Differences observed when using Image1 as reference to describe Image2.)*
> *[2] (Differences observed when using Image2 as reference to describe Image1.)*
> Output only the two lines above and nothing else.

Consequently, each image pair yields two triplet annotations, resulting in 600 triplets that are used to train IDPG.

### A.2    DETAILS OF THE TIPS FRAMEWORK

The complete workflow and detailed instructions of our automated Text-Image Pairs Synthesis (TIPS) framework are illustrated in Figure A1, consisting of three stages: diversified prompt generation driven by Large Language Model (LLM) Yang et al. (2025), identity-preserving person image generation, and image quality filtering with caption generation. Each component is described in detail in the following subsections.

#### DIVERSIFIED PROMPT GENERATION

In zero-shot scenarios, we strive to generate person prompts with sufficient diversity and minimal repetition. Therefore, we design the instructions as depicted in Figure A1, containing three critical random elements: the suggested character, color, and clothing. These elements are randomly selected from pre-defined lists, and the resulting sentences are explicitly required to include the recommended descriptive elements to enhance the coverage and diversity of the generated prompts. However, extensive generation inevitably leads to identical combinations, causing similar prompt outputs. To mitigate this, the instructions also incorporate three randomly selected examples from SPG's training data, ensuring output stability and maximizing textual diversity simultaneously.

In few-shot and generalization scenarios, to better approximate the style of manual annotations, we redesign the prompt generation instructions. Three reference sentences are randomly selected from

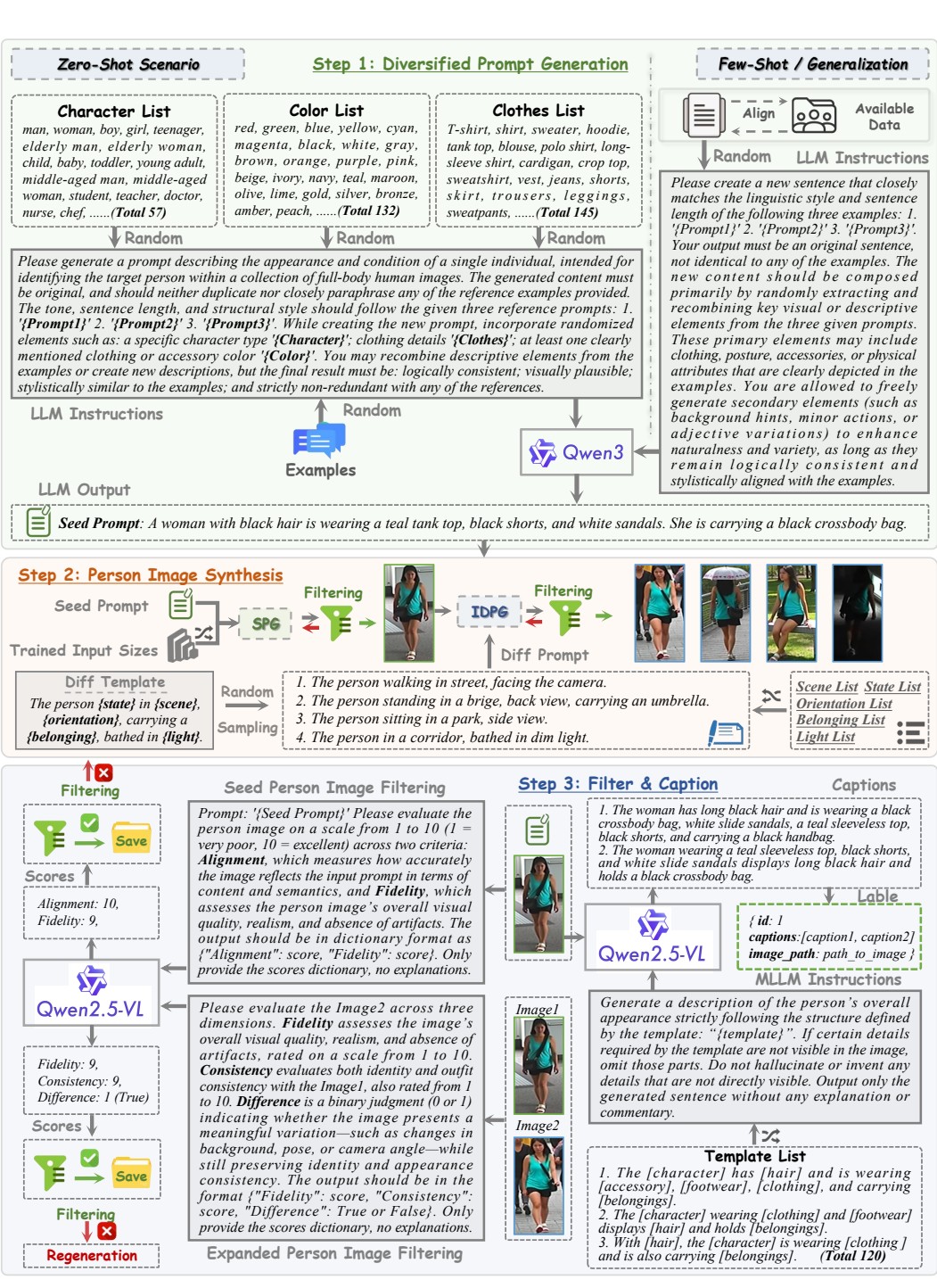

Figure A1: Pipeline of TIPS framework with detailed instruction design.

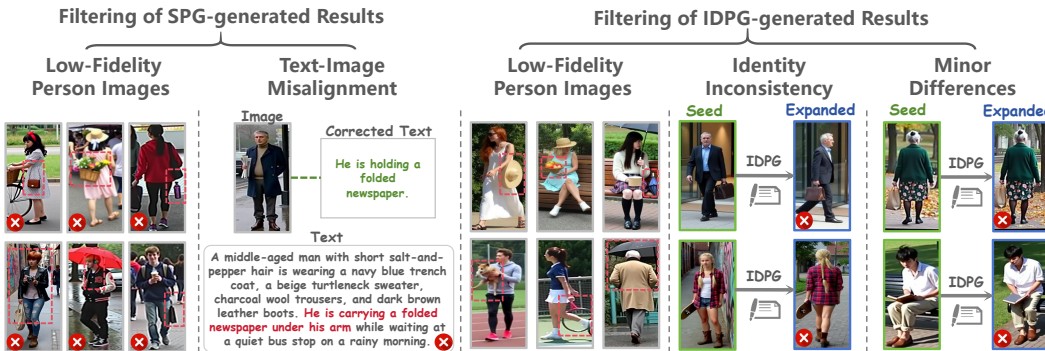

Figure A2: Representative examples of samples filtered out during the data selection process. From left to right, each panel corresponds respectively to two evaluation dimensions for SPG and three for IDPG. Samples shown here are excluded due to low scores in their corresponding evaluation dimensions. Red boxes highlight issues related to image naturalness and fidelity.

existing texts to guide sentence length and style. The LLM is then tasked to form new sentences using elements appearing in these three reference sentences. This design not only helps the language style approximate manual annotations but also stabilizes SPG's generation and aligns the outputs closely with the training domain distribution since all elements provided to the LLM have previously appeared during SPG training. With the above strategies, we can automatically generate a large quantity of qualified textual prompts suitable for various scenarios.

IDENTITY-PRESERVING PERSON IMAGE SYNTHESIS

After generating prompts, the corresponding SPG and IDPG trained for each scenario are used to synthesize person images. Initially, SPG generates a seed person image for each prompt. Although the physical resolution of generated images is fixed at $192 \times 384$, we simulate real-world multi-resolution scenarios by adjusting the image size conditions, thereby achieving "multi-resolution" seed person image generation with varying clarity levels. The resolution conditions used for image generation are randomly selected from a pre-defined size list employed during SPG training, ensuring stable and consistent image quality. All seed images undergo MLLM filtering after generation, and any images failing the filtering criteria are regenerated.

After seed image generation, a single image corresponding to a new identity is obtained. The responsibility of IDPG is to expand this single image into an image set of the same identity. Through training, IDPG acquires the ability to generate identity-preserving target images, where a reference image and relative textual differences serve as inputs, and the generated outputs match both the identity of the reference image and the provided textual differences. Therefore, the seed person image serves as the reference image, and four relative textual differences are randomly generated for each seed image from multiple pre-defined lists, using the difference-text template illustrated in Figure A1. These relative texts and the reference seed image are then fed into IDPG to produce four expanded images. These images must also pass MLLM filtering criteria, and any images failing to meet these conditions are regenerated. Thus, we ultimately obtain five person images sharing the same identity.

FILTER AND CAPTION

After obtaining five images of the same identity, corresponding textual annotations are generated to form image–text pairs suitable for training Text-based Person Retrieval (TPR) models. We next detail the MLLM-based filtering criteria. The filtering of seed person images generated by SPG relies on two dimensions, as illustrated in Figure A1. The first evaluates the alignment between the generated seed images and their textual prompts; higher alignment scores indicate that the generated content faithfully reflects the intended description. The second dimension assesses intrinsic image quality, ensuring that the generated person images exhibit high fidelity, natural realism, and minimal artifacts. Both dimensions are scored on a 1–10 scale, and only images achieving scores of 9 or above in both criteria pass the filtering step, thereby guaranteeing high-quality seed images. Under

the zero-shot setting, the overall rejection rate is 19.6% (15.5% low-fidelity images, 4.6% text–image misalignment, and 0.5% both), compared with 17.5% under the few-shot setting (14.3% low-fidelity, 3.6% misalignment, 0.4% both) and 14.6% under the generalization setting (11.9% low-fidelity, 3.1% misalignment, 0.4% both).

For expanded images generated by IDPG, three filtering criteria are applied to each image. The inputs include the seed image and its corresponding expanded version. The first criterion again evaluates image quality, consistent with the SPG filtering step. The second checks identity consistency between the expanded and reference images, ensuring that crucial identity information is retained. The third criterion assesses variability, requiring the expanded image to exhibit noticeable differences from the seed image. The first two dimensions use the same 1–10 scoring scale, with scores required to reach at least 9. The third employs binary classification, where the MLLM determines whether the expanded image displays sufficient distinction. Images failing this requirement are discarded. The rejection rate for expanded images remains relatively stable across scenarios, averaging 36.2%, with 23.0% attributed to low-fidelity images, 10.5% to identity inconsistency, and 3.9% to insufficient variation. This strict multi-dimensional filtering significantly suppresses identity drift in the final dataset. Figure A2 provides visual examples of SPG- and IDPG-generated person images rejected under different criteria, illustrating the effectiveness of the filtering strategy. Figure A8 shows additional IDPG-generated images that were rejected despite closely approaching the identity-consistency threshold, further highlighting the strictness of the filtering process.

After filtering, two distinct captions are generated for each of the five retained images belonging to the same identity. As illustrated in Figure A1, this process involves randomly selecting two templates from a predefined set of 120. Each template is independently inserted into an instruction and provided to the MLLM, resulting in two distinct captions for each image. The template set ensures structural diversity in the final captions, thereby improving the robustness of retrieval models trained on the textual modality.

## B  THEORETICAL FEASIBILITY ANALYSIS

**Proposition.**   Let $\mathcal{H}$ be a hypothesis space for a dual-encoder retrieval model that embeds an image $x$ and a text $y$ into a common metric space and returns a similarity score $h(x, y)$. Denote by $P_s$ the joint distribution of *synthetic* image-text pairs used for pre-training and by $P_t$ the joint distribution of *target-domain* pairs on which the model is evaluated. Assuming each pair $(x, y) \sim P_s \cup P_t$ is *aligned* (the text truly describes the image), the smaller the divergence $D(P_s, P_t)$ between the two distributions, the lower the expected retrieval error on the target domain.

**Proof.**   Define the *binary retrieval loss*:

$$\ell_h(x, y, x', y') = \mathbb{1}\!\!\!\!\nmid [h(x, y) < h(x', y')], \tag{6}$$

which equals 1 when a negative pair $(x, y)$ is scored higher than a positive pair $(x', y')$, and 0 otherwise. Writing

$$\epsilon_s(h) = \mathbb{E}_{P_s^+ \times P_s^-}[\ell_h], \quad \epsilon_t(h) = \mathbb{E}_{P_t^+ \times P_t^-}[\ell_h], \tag{7}$$

where $P^+$ and $P^-$ denote positive and negative pair distributions under $P$, we invoke the standard domain-adaptation decomposition:

$$\epsilon_t(h) \leq \epsilon_s(h) + D(P_s, P_t) + \lambda_*, \tag{8}$$

where $D(\cdot, \cdot)$ is any symmetric discrepancy measure (e.g., Wasserstein, total variation, or $\mathcal{H}\Delta\mathcal{H}$-divergence), and $\lambda_* = \min_{h' \in \mathcal{H}}[\epsilon_s(h') + \epsilon_t(h')]$ is an irreducible error term determined solely by the hypothesis class.

The alignment assumption guarantees that, for every $h$, the source risk $\epsilon_s(h)$ can be driven arbitrarily close to $\epsilon_{\text{bayes}}$ (the Bayes error) through sufficient training, since misleading image-text mismatches are absent. Consequently, we have:

$$\epsilon_s(h) = \epsilon_{\text{bayes}} + \delta_s, \quad 0 \leq \delta_s \ll 1. \tag{9}$$

Similarly, since both domains share the same label semantics, $\lambda_*$ is lower-bounded by the same $\epsilon_{\text{bayes}}$ and thus behaves as a constant with respect to $D(P_s, P_t)$. Substituting into equation 8 yields:

$$\epsilon_t(h) \leq \epsilon_{\text{bayes}} + \delta_s + D(P_s, P_t) + \lambda_* - \epsilon_{\text{bayes}}. \tag{10}$$

Collecting constants results in the bound:

$$\epsilon_t(h) \leq C + D(P_s, P_t), \tag{11}$$

where $C = \delta_s + \lambda_* - \epsilon_{\text{bayes}}$ is independent of $D(P_s, P_t)$. Inequality equation 11 demonstrates that the target retrieval error increases at most linearly with the distribution discrepancy. Therefore, under fixed image-text alignment, *reducing $D(P_s, P_t)$*, i.e., making the synthetic image distribution more similar to the target-domain distribution, *strictly tightens* the generalization bound and thus improves expected retrieval performance.

**Application.**  The above theoretical proof serves as the starting point and a critical objective for synthetic data generation in this paper. Initially, we utilize advanced generative models along with MLLM-based filtering and captioning to ensure a high degree of alignment between generated images and their corresponding texts. On this basis, we further fine-tune the person image generators, enabling the generated images to closely approximate the target-domain distribution, thereby achieving superior retrieval performance.

## C  ADDITIONAL DETAILS

### C.1  DATASETS DETAILS

**CUHK-PEDES** Li et al. (2017) (CUHK) serves as a foundational benchmark for text-to-person retrieval, containing 40,206 images and 80,412 manually annotated textual descriptions across 13,003 unique identities. The dataset is formally partitioned into three subsets: a training set with 34,504 images and 68,126 descriptions covering 11,003 identities, a validation set comprising 3,078 images and 6,158 descriptions for 1,000 identities, and a test set of 3,074 images paired with 6,156 descriptions representing another 1,000 identities. Each image is associated with two independent textual annotations, with an average description length exceeding 23 words to ensure comprehensive semantic coverage.

**ICFG-PEDES** Ding et al. (2021) (ICFG) offers 54,522 precisely aligned image-text pairs spanning 4,102 identities, distinguished by its single-description-per-image annotation strategy. The textual component demonstrates lexical richness through 5,554 unique vocabulary terms, with descriptions averaging 37 words for detailed attribute specification. Dataset division yields 34,674 training pairs across 3,102 identities and 19,848 test pairs for the remaining 1,000 identities, emphasizing granular identity representation through text-visual correspondence.

**RSTPReid** Zhu et al. (2021) (RSTP) addresses practical surveillance challenges through multi-camera acquisition, containing 20,505 images and 41,010 textual descriptions for 4,101 identities captured across 15 viewpoints. Each identity features five cross-view images accompanied by dual descriptions, all maintaining a minimum length of 23 words. The dataset follows a structured partitioning scheme with 3,701 identities for training, while both validation and test sets contain 200 identities each, facilitating rigorous evaluation under real-world deployment conditions.

### C.2  METHOD EFFICIENCY

Our proposed method, which can be adapted to virtually all existing TPR methods, introduces additional time overhead primarily in two aspects. The first overhead is associated with data preparation, a one-time cost per scenario, after which the generated data can be repeatedly utilized for training multiple feasible models. The second overhead occurs during inference when optionally enabling Test-Time Augmentation (TTA) for performance enhancement. Using the hardware configuration of two H800 GPUs with 80GB memory each and the settings described in Section 4.1, the training of the SPG requires approximately 4 hours and 19 minutes per scenario, while training the general IDPG takes approximately 6 hours and 43 minutes. After generator training, the TIPS data expansion framework generates 400,000 image-text pairs per scenario within 134 hours and 58 minutes, averaging 2.43 seconds per sample (single GPU), significantly improving efficiency compared to manual annotation.

For the TPR task, inference efficiency of the model is of greater importance. When TTA is disabled, our approach maintains the original inference efficiency of the baseline models while improving

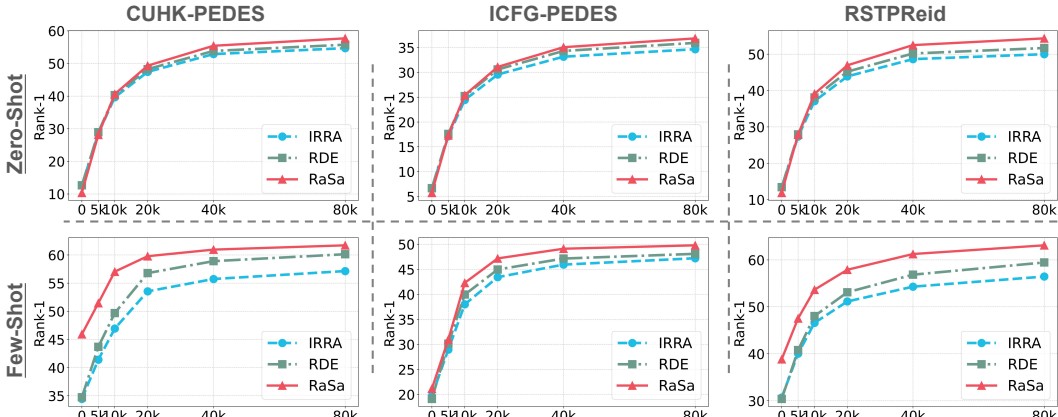

Figure A3: Impact of the Amount of Data Expansion via the TIPS Framework on Zero-shot and Few-shot TPR Performance. The top three plots correspond to the zero-shot setting, while the bottom three correspond to the few-shot setting. From left to right, the three columns show performance variation curves on the CUHK-PEDES, ICFG-PEDES, and RSTPReid datasets, respectively. In each subplot, the X-axis indicates the number of text prompts generated by the TIPS framework, with the corresponding number of image-text pairs being ten times that value.

retrieval performance since modifications are restricted solely to training data. With TTA enabled, an additional average text-processing time of 2.75 seconds per query is required to generate preview images. After completing preview generation, we perform multiple single-GPU inference evaluations for three representative methods (IRRA Jiang & Ye (2023), RDE Qin et al. (2024), and RaSa Bai et al. (2023)) on the full CUHK-PEDES test set (consisting of 6,156 textual queries and 3,074 candidate images) to minimize the impact of random variance on inference time measurement. For methods without re-ranking, TTA significantly impacts efficiency due to additional visual feature computation and fusion processes: inference time increases from 5.34 seconds to 11.18 seconds for IRRA, and from 11.22 seconds to 21.45 seconds for RDE. For the re-ranking-based RaSa method, whose computational overhead primarily concentrates in the re-ranking stage, enabling TTA increases inference time only marginally from 510.37 seconds to 516.61 seconds, representing a modest overhead of approximately 1.22%. This indicates that users can flexibly activate TTA according to their specific trade-off requirements between performance and efficiency.

## C.3 IMPLEMENTATION DETAILS.

In the experiments, we simulate three realistic scenarios. The first one is the zero-shot scenario, where no corresponding image-text pair annotations exist for the new domain. The second one is the few-shot scenario, in which only a minimal number of samples are available. The last one is the generalization scenario, where the annotated data used do not correspond directly to the current domain.

To handle the zero-shot scenario, we uniformly select 100 IDs from five datasets (CUHK03 Li et al. (2014), CUHK02 Li & Wang (2013), Market-1501 Zheng et al. (2015), MSMT17 Wei et al. (2018), and VIPER Gray & Tao (2008)), and these IDs do not overlap with the image sources of the TPR artificial dataset. From each ID, we randomly select three images, amounting to 300 images in total, and then employ MLLM to generate two captions for each image so as to train the SPG. Image pairs from the same ID but different images are used as reference images, and their differential captions are used to train the IDPG. Note that in scenarios where annotated data is available, SPG's training utilizes the provided image-text pairs.

All experiments are conducted using two H800 GPUs. We select Qwen3-32B Yang et al. (2025) as our LLM and Qwen2.5VL-32B Bai et al. (2025) as the MLLM. Both SPG and IDPG are based on FLUX.1-dev Labs (2024), and the LoRA rank is set to $r = 32$. During training, each GPU employs a batch size of 1, with a 2-step gradient accumulation, using the AdamW optimizer Loshchilov & Hutter (2017) (learning rate $1 \times 10^{-5}$, 10-step warmup, weight decay of 0.01) for a total of 20,000 steps. All input images are resized to $192 \times 384$. During SPG training, width-height resolution

conditions are randomly dropped in 20% of samples to enhance the robustness. For IDPG training, difference captions are dropped with a probability of 10% to strengthen the model's ability to generate different images of the same identity by default. For each experiment in each scenario, the LLM generates 40,000 independent seed prompts. Each text is paired with randomly sampled resolution parameters trained by the SPG to generate one image of size $192 \times 384$ through 28 sampling steps. These images are then fed into the IDPG, generating four additional images of the same identity using four random difference prompts. The generated images and their seed image share the same ID. All images undergo MLLM filtering, where the scoring rules for each evaluation dimension require a binary classification to determine whether the differences exist between seed and expanded images, and scores from 1 to 10 for all other dimensions, requiring a minimum score of 9. Images failing to meet these criteria are regenerated. Subsequently, all filtered images are provided two captions generated by MLLM using randomly selected templates. Therefore, each experiment will yield 40,000 IDs, 200,000 images, and 400,000 image-text pairs for retrieval model training.

Three representative TPR methods (IRRA Jiang & Ye (2023), RDE Qin et al. (2024), Rasa Bai et al. (2023)) are evaluated on CUHK, ICFG, and RSTP datasets, following their original configurations. Training in the few-shot scenario comprises two phases: 1) synthetic data training with original hyperparameters, 2) fine-tuning with real data, with epochs and learning rates halved. In TTA, the value of $\alpha$ is set to 0.6. For IRRA, the ID loss Zheng et al. (2020) layer parameters from *Phase 1* are excluded in *Phase 2*, but other parameters are retained.

# D  ADDITIONAL RESULTS

## D.1  ADDITIONAL ABLATION STUDIES

### IMPACT OF EXPANDED DATA QUANTITY IN TIPS

Figure A3 illustrates how varying the quantity of expanded data influences retrieval performance under both zero-shot and few-shot settings. Under both scenarios and across each dataset, a consistent trend is observed: as the number of text prompts expanded by TIPS increases, the retrieval performance improves gradually, with the rate of improvement diminishing as the quantity continues to increase. Specifically, retrieval performance rises rapidly until the number of expanded prompts reaches approximately 20,000. Beyond this threshold, the performance continues

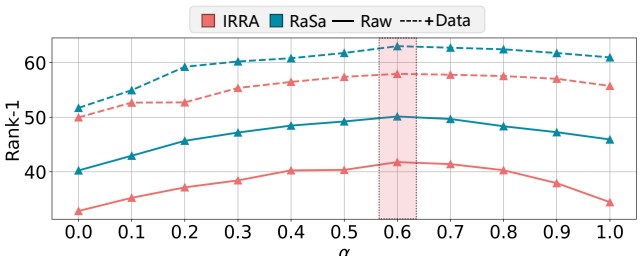

Figure A4: Retrieval performance of different methods with TTA using varying $\alpha$ values under the few-shot scenario on the CUHK dataset. Optimal $\alpha$ values are highlighted in red boxes.

to improve, albeit at a significantly reduced pace. Ultimately, to balance performance and efficiency, we choose to expand 40,000 prompts for each scenario using the TIPS framework, resulting in a total of 400,000 trainable image-text pair samples.

### TTA HYPERPARAMETER $\alpha$

Figure A4 analyzes the impact of the TTA hyperparameter $\alpha$ on retrieval performance under the few-shot setting using the CUHK dataset. When $\alpha = 0$, retrieval is conducted solely based on preview images generated from the textual queries in the test set. It is evident that even in this scenario, the model achieves reasonable performance without specific optimization, establishing a prerequisite condition for the effectiveness of TTA. To fully leverage the capability of TTA, it is necessary to identify the optimal balance between textual retrieval and preview-image-based retrieval. As demonstrated in the figure, the optimal hyperparameter under the current experimental setup is found to be $\alpha = 0.6$, which is consequently adopted as a general parameter setting in the few-shot scenario. In practical applications, regardless of the specific scenario, since TTA does not require any modification to the training procedure, the optimal value of $\alpha$ can be rapidly determined through a grid

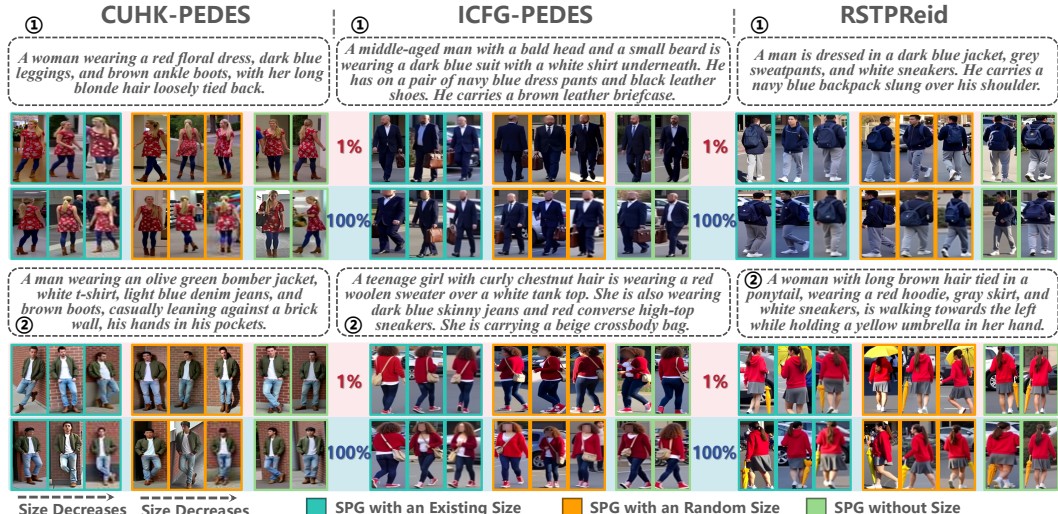

Figure A6: Examples generated by SPG using different resolution conditions and textual inputs. Each column presents two sample groups from each dataset. Within each group, the first row shows images generated by the model trained with 1% of the data, and the second row shows images generated by the model trained with the full dataset. Each case includes three subsets generated with the same textual input but under different resolution conditions, from left to right: a resolution seen during training, a randomly sampled untrained resolution, and no resolution condition.

search on a validation set. Thus, TTA can be effectively activated to achieve stable performance improvements when ultimate retrieval performance is desired.

IMPACT OF LoRA RANK CONFIGURATIONS

To determine the optimal rank values of LoRA utilized in the SPG and IDPG, we evaluate their performance under various rank settings on the CUHK-PEDES dataset within a few-shot scenario. Here, the Fréchet Inception Distance (FID) metric quantifies the distributional divergence between expanded images and the CUHK-PEDES test set. Conducting a direct binary search jointly on the rank values of SPG and IDPG would require substantial computational resources, as each evaluation involves gener-

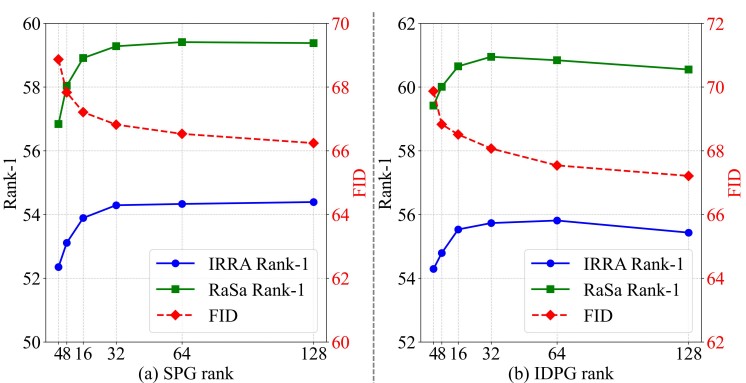

Figure A5: SPG and IDPG LoRA rank impact on retrieval performance and expanded data distribution. (a) effect of SPG rank settings on performance; (b) effect of IDPG rank settings on performance with SPG rank fixed at 32.

ating 400,000 image-text pairs. Therefore, we first optimize the rank value for the SPG individually and subsequently determine the optimal rank for the IDPG based on this result. In the scenario involving only SPG, the TIPS framework directly employs the SPG to generate five images per prompt, representing images of the same identity. Figure A5a illustrates that as the rank increases, SPG's number of learnable parameters grows, enhancing its fitting ability to the training-domain distribution, and thus steadily reducing FID scores, signifying improved alignment with the target domain. However, this improvement at the distributional level does not linearly translate into better retrieval performance. Specifically, retrieval metrics improve when the rank is below 32, yet the

Table A1: Performance comparison of different resolution control strategies in SPG.

| Method | Zero-Shot | | Few-Shot | |
|---|---|---|---|---|
| | R@1 | mAP | R@1 | mAP |
| Baseline | – | – | 34.44 | 32.57 |
| w/o size | 51.23 | 46.85 | 54.04 | 48.79 |
| Low | 51.93 | 47.22 | 54.90 | 49.36 |
| High | 50.18 | 45.92 | 53.15 | 47.64 |
| **Ours** | **52.89** | **47.81** | **55.73** | **49.72** |

benefits diminish beyond this point. Notably, a slight performance degradation is observed when the rank surpasses 64, likely due to overfitting, wherein an excessive number of parameters captures domain-specific artifacts, consequently reducing generation diversity. Conversely, insufficient ranks (e.g., ranks below 32) limit the model's capacity to adequately learn domain characteristics. Balancing efficacy and efficiency, we thus select rank $r = 32$ for SPG as the final configuration.

Based on this setting, we incorporate IDPG to expand person images, with the performance under different ranks illustrated in Figure A5b. Observing the trends, we find a similar phenomenon to SPG: increasing rank values progressively reduce FID scores between generated images and the test domain. Nevertheless, when the rank exceeds 32, the introduction of additional trainable parameters leads to a decrease in retrieval performance. This occurs because the abundance of parameters encourages the IDPG to preserve not only the original person characteristics but also excessive background details. Given that SPG has been trained on limited samples from the current domain, excessive imitation of backgrounds effectively reduces FID but results in more monotonous expanded images, thereby negatively affecting retrieval performance. Considering these factors comprehensively, we similarly adopt rank $r = 32$ for IDPG as the final configuration.

### RESOLUTION CONTROL AND IMAGE QUALITY

To provide a comprehensive assessment of how pixel-level resolution control influences retrieval performance, we conduct systematic experiments evaluating several resolution conditioning strategies within our TIPS framework. Using IRRA as the baseline model on the CUHK dataset under both zero-shot and few-shot settings, we compare four distinct resolution control variants to validate the effectiveness of our design.

The experimental settings are defined as follows: **w/o size** removes all resolution conditioning during SPG inference, yielding images without explicit size guidance; **Low** uniformly applies the lowest resolution used during training; **High** enforces the highest training-time resolution; and **Ours** adopts our strategy of sampling resolution conditions randomly from the empirical training distribution during inference.

As shown in Table A1, our random resolution conditioning strategy achieves the best performance across both evaluation regimes, obtaining 52.89% zero-shot R@1 and 55.73% few-shot R@1. Importantly, the **Low** setting consistently outperforms the **High** setting in both zero-shot and few-shot scenarios by 1.75% in R@1. This suggests that high-resolution conditioning may steer the generator toward overly clean synthetic distributions, thereby reducing generalization when tested on real-world data exhibiting diverse and often degraded resolutions. Conversely, low-resolution conditioning imposes a more challenging generation objective, preventing over-specialization and enhancing robustness.

Furthermore, our method demonstrates clear improvements over the no-conditioning baseline (**w/o size**), with gains of 1.66% in zero-shot R@1 and 1.69% in few-shot R@1. These results highlight the importance of explicit resolution modeling in synthetic data generation. By sampling resolutions from the training distribution, our approach effectively balances learning difficulty and generalization capacity, faithfully capturing the multi-resolution characteristics commonly found in practical TPR scenarios while maintaining controlled generation behavior.

Overall, these findings confirm that pixel-level resolution control is a critical component of the SPG design, substantially contributing to its ability to generate domain-adaptive person images and improving retrieval performance across diverse real-world conditions.

Table A2: Performance comparison between data reuse and regeneration strategies in few-shot scenarios.

| Pre-training Data | CUHK | | ICFG | | RSTP | |
|---|---|---|---|---|---|---|
| | R@1 | mAP | R@1 | mAP | R@1 | mAP |
| No Pre-training | 34.44 | 32.57 | 19.73 | 10.20 | 30.70 | 24.65 |
| Ours (Reuse) | 53.28 | 46.98 | 40.59 | 21.73 | 51.15 | 40.03 |
| Ours (Regeneration) | **55.73** | **49.72** | **45.94** | **24.75** | **54.30** | **42.00** |

Table A3: Performance comparison across different LLM scales for prompt generation.

| LLM | Zero-Shot | | Few-Shot | |
|---|---|---|---|---|
| | R@1 | mAP | R@1 | mAP |
| Baseline | – | – | 34.44 | 32.57 |
| Qwen3-8B | 51.95 | 47.29 | 54.99 | 48.32 |
| Qwen3-14B | 52.53 | 47.76 | 55.60 | 49.18 |
| **Ours (Qwen3-32B)** | **52.89** | **47.81** | **55.73** | **49.72** |

## COMPUTATIONAL EFFICIENCY AND DATA REUSABILITY

Data reusability offers a practical pathway toward improved computational efficiency. As shown in Table A2, directly reusing the zero-shot expanded dataset without regeneration (**Ours (Reuse)**) already provides substantial performance gains across all few-shot benchmarks, achieving 53.28% R@1 on CUHK, 40.59% on ICFG, and 51.15% on RSTP. This corresponds to an average improvement of 18.84% R@1 over the no pre-training baseline, while entirely avoiding the computational overhead associated with data regeneration.

For applications requiring peak performance, regeneration (**Ours (Regeneration)**) leads to additional improvements of 2.45% on CUHK, 5.35% on ICFG, and 3.15% on RSTP. These gains arise because retraining the generators allows the synthetic data distribution to better align with the target domain.

These findings indicate that although the initial data synthesis stage requires computational investment, the resulting synthetic datasets possess long-term reusability and scalability.

## LLM SCALE AND ROBUSTNESS ANALYSIS

To further investigate the influence of language model capacity on our TIPS framework, we conduct systematic experiments examining how different LLM scales affect downstream retrieval performance when generating textual prompts. Using IRRA as the baseline model on the CUHK dataset under both zero-shot and few-shot settings, we compare three variants of the Qwen3 Yang et al. (2025) model family with varying parameter counts.

As shown in Table A3, replacing our default LLM with smaller variants leads to only marginal reductions in retrieval performance. The Qwen3-8B model achieves 51.95% zero-shot R@1 compared with 52.89% obtained using Qwen3-32B, representing a negligible decrease of 0.94%. A similar trend is observed in few-shot settings, where Qwen3-8B yields 54.99% R@1 versus 55.73% with the full-scale model. These results indicate that the LLM's primary role within our framework is to produce semantically coherent and stylistically diverse prompts—a task that smaller models can accomplish effectively.

The observed robustness arises from two key design aspects. First, the diversity in prompt generation is primarily driven by our template-based and randomized instruction design rather than by the inherent generative capacity of the LLM. The incorporation of high-quality exemplars in the instruction prompts ensures that even smaller LLMs produce well-formed and structurally sound outputs suitable for image generation. Second, the subsequent MLLM-based filtering stage effectively mitigates limitations of weaker LLMs by removing low-quality or misaligned prompts, preventing errors from propagating to later stages of the pipeline.

Table A4: Comprehensive ablation study on MLLM filtering and captioning strategies. "Similarity" measures stylistic similarity (lower is better, scale 1–10).

| No. | Method | Zero-Shot | | | Few-Shot | | |
|-----|--------|-----------|-----|-------------|----------|-----|-------------|
| | | R@1 | mAP | Similarity ↓ | R@1 | mAP | Similarity ↓ |
| 1 | **Baseline** | – | – | – | 34.44 | 32.57 | – |
| 2 | **w/o Filtering** | 49.45 | 45.26 | **4.75** | 52.13 | 47.75 | **5.01** |
| 3 | **Simple Filtering** | 50.73 | 46.23 | 4.79 | 53.54 | 48.31 | 5.09 |
| 4 | **w/o Template** | 47.55 | 43.88 | 6.59 | 51.80 | 47.09 | 6.93 |
| 5 | **30 templates** | 50.67 | 46.31 | 5.46 | 53.35 | 47.93 | 5.59 |
| 6 | **Ours (Full)** | **52.89** | **47.81** | 4.82 | **55.73** | **49.72** | 5.03 |

These experiments collectively demonstrate that our framework exhibits strong robustness to variations in LLM scale, with differences across model sizes generally remaining within 1%. The method does not rely heavily on large-scale language models or their raw generative capabilities, as stylistic variation at the prompt level is reliably regulated by template-based diversification and downstream quality control. This property has practical implications for resource-constrained deployments, in which smaller LLMs can be adopted without incurring meaningful performance degradation, thereby substantially reducing computational costs.

MLLM FILTERING AND CAPTIONING STRATEGIES

To comprehensively evaluate the impact of MLLM-based filtering and captioning strategies on downstream TPR performance, we conduct extensive ablation studies using IRRA as the baseline model on the CUHK dataset under both zero-shot and few-shot settings. The experimental results, presented in Table A4, systematically analyze the effects of different filtering strictness levels and the influence of template diversity in caption generation.

The ablation settings are defined as follows: **w/o Filtering** removes all MLLM-based quality assessment, retaining all generated images; **Simple Filtering** replaces our full multi-dimensional evaluation criteria with drastically simplified binary classification prompts, using the following instructions strictly without modification: *"Prompt: '{Seed Prompt}' Evaluate the person image considering Alignment (image-text similarity) and Fidelity (visual quality of the person image). Output only 'yes' or 'no' as the final overall result."* for SPG, and *"Evaluate Image2 for Fidelity (visual quality of the person image), Consistency (identity and outfit consistency with Image1), and Difference (whether there's a meaningful variation while preserving identity). Output only 'yes' or 'no' based on the overall assessment."* for IDPG. **w/o Template** removes template guidance during captioning, allowing free-form MLLM descriptions; **30 templates** uses a reduced template set; and **Ours** denotes the complete TIPS framework with 120 templates and full multi-dimensional filtering, using strict acceptance thresholds where all 10-point scoring dimensions must score $\geq 9$ and all binary criteria must pass.

Our results show that MLLM filtering is essential for ensuring synthetic data quality, yielding more than 3% Rank-1 improvement in both zero-shot and few-shot settings relative to unfiltered data. Replacing our detailed multi-dimensional filtering with simple binary classification leads to a 2.16% drop in zero-shot R@1, confirming that our stringent filtering criteria effectively constrain MLLM bias and prevent noisy samples from degrading the training set.

To quantify stylistic similarity, we employ Qwen2.5-VL-32B Bai et al. (2025) with strictly preserved instructions: *"Given two sentences, evaluate the stylistic similarity between them. Focus on factors like sentence length, complexity, use of formal or informal language, word choice, and overall flow. Do not focus on the semantic content, but rather on how similar the style and structure are between the two sentences. Provide a score between 1 and 10."* This evaluator focuses exclusively on stylistic rather than semantic attributes. Scores are averaged over 5000 randomly sampled sentence pairs for each condition, ensuring statistical reliability. Identical image sets are used across captioning variants to isolate the impact of textual changes.

Template-free captioning results in substantially higher stylistic similarity (6.59 in zero-shot), indicating more homogeneous writing patterns that weaken model robustness. Increasing template diversity consistently improves performance, with our 120-template setup achieving the lowest stylistic

similarity (4.82) and the highest retrieval accuracy. This result demonstrates that template-guided captioning effectively mitigates MLLM stylistic bias while enhancing linguistic diversity consistent with real-world human annotations.

Overall, these comprehensive ablations confirm that both our multi-stage filtering mechanism and template-based captioning strategy are indispensable components of the framework. They work synergistically to produce high-quality synthetic data, suppress potential MLLM biases, and maximize downstream TPR performance.

## D.2 QUALITATIVE RESULTS

### MULTI-RESOLUTION GENERATION OF SPG

Figure A6 visually demonstrates how the SPG, trained under different data scales, effectively preserves the inherent characteristics of each dataset while generating multi-resolution images. By examining these images, we observe that synthetic samples (highlighted in blue) accurately reproduce the resolution distributions and stylistic attributes of the original images across various data scenarios. Specifically, the expanded samples from CUHK retain its broad resolution distribution, including very low-resolution images, whereas the ICFG and RSTP datasets maintain their intrinsic clarity characteristics. Under conditions involving resolutions that were not seen during training (highlighted in orange), SPG trained on extensive data still effectively generates results of varying clarity based on different resolution conditions. Conversely, models trained on limited data fail to demonstrate robust zero-shot resolution adaptability, resulting in generated images that inadequately reflect the intended resolution inputs due to insufficient training across a wide range of resolutions. Therefore, within the TIPS framework, to ensure stable resolution control during seed person image generation, resolutions are consistently sampled from a list of resolutions encountered during training. Additionally, in cases

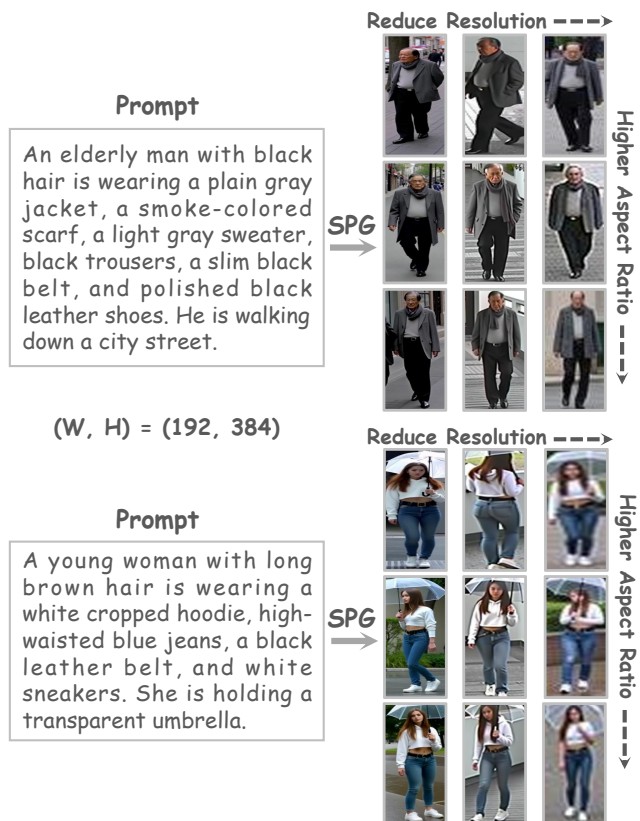

Figure A7: Images generated by SPG with identical prompts but different resolution conditions under a fixed physical resolution setting. Within each group, image clarity decreases from left to right as the resolution lowers, while the aspect ratio increases from top to bottom.

without explicit resolution conditions (highlighted in green), generated images default to a preferred degree of blur specific to each model. Moreover, the trained SPG successfully retains inherent dataset-specific attributes, such as the characteristic pixelation of facial regions in the ICFG dataset, allowing rapid expansion of additional images conforming to the original dataset distribution even under low-data scenarios. Figure A7 further illustrates the precise control offered by SPG's resolution conditioning, which accurately modulates image clarity and aspect ratios at fixed physical resolutions.

In particular, the textual inputs shown in Figure A6 are also generated by the LLM under a few-shot scenario. Observations confirm that the generated texts successfully emulate the unique linguistic styles of each dataset, including complete sentence descriptions typical of CUHK and age-prefixed,

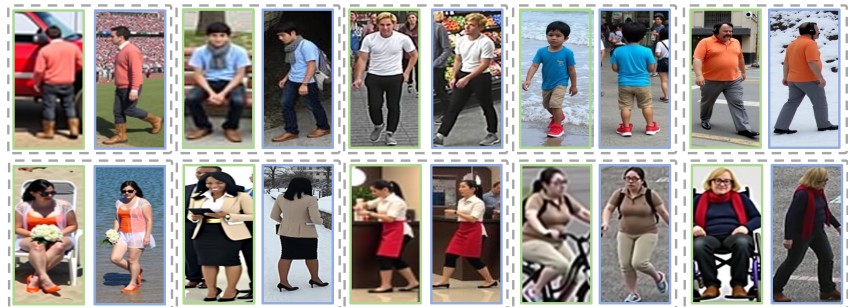

Figure A8: Ten data groups with an MLLM identity-consistency score of 8. For each group, the left image shows the seed character generated by SPG, and the right image shows the extended character generated by IDPG.

long-form annotations seen in ICFG. More importantly, the LLM introduces entirely new scenarios and clothing combinations absent from the original data, significantly enhancing data diversity during the seed image generation phase of the TIPS framework. When further combined with IDPG, image filtering, and caption generation, this approach enables the creation of higher-quality and more diverse training data for TPR.

# E    EXPANDED DATA DETAILS

This section provides a comprehensive analysis and visualization of the data expanded by our TIPS framework, encompassing statistical evaluation of textual diversity, quantitative assessment of visual consistency and variation, and representative visual demonstrations across different experimental settings.

## E.1    STATISTICAL ANALYSIS OF GENERATED CAPTIONS

Quantitative analysis of the generated captions demonstrates superior linguistic diversity compared with manually annotated TPR datasets. The zero-shot expanded dataset achieves a vocabulary size of 10,432 unique words and an average sentence length of 35.84 words, substantially exceeding the corresponding statistics for CUHK (7,147 words, 23.82 words), ICFG (3,005 words, 35.58 words), and RSTP (2,903 words, 26.53 words). This lexical richness is further reflected in the number of unique words per sentence, with an average of 30.09 for the zero-shot dataset versus 19.07–26.17 in real-world datasets, indicating that our template-guided MLLM captioning produces more diverse textual descriptions than human annotations.

In the few-shot scenario, textual diversity is moderately reduced, with a vocabulary size of 7,768 words and an average sentence length of 30.71 words. This decrease results from the closer distributional alignment between generated seed person images and the target domain, which limits visual variability and consequently reduces lexical diversity in the MLLM-generated captions. Despite this domain-adaptive specialization, the few-shot captions still maintain greater linguistic richness than manually annotated datasets, confirming the robustness and effectiveness of our template-based captioning strategy across diverse application contexts.

## E.2    QUANTITATIVE ANALYSIS OF DATASET DIVERSITY AND IDENTITY CONSISTENCY

To comprehensively evaluate the diversity and identity consistency of our generated dataset, we conduct rigorous quantitative analyses using specialized metrics and established models. For identity consistency assessment, we employ SOLIDER Chen et al. (2023) (Swin-Base Liu et al. (2021)) trained on Market1501 Zheng et al. (2015) to extract global embeddings and compute cosine similarity between image pairs, as traditional face recognition methods are unreliable for TPR datasets due to frequent facial occlusions and blurring. The results show that the IDPG effectively preserves identity, achieving a similarity score of 0.919, closely matching real-data baselines of 0.924 on ICFG and 0.916 on RSTP test sets. This substantially outperforms SPG-only generation (0.886 similar-

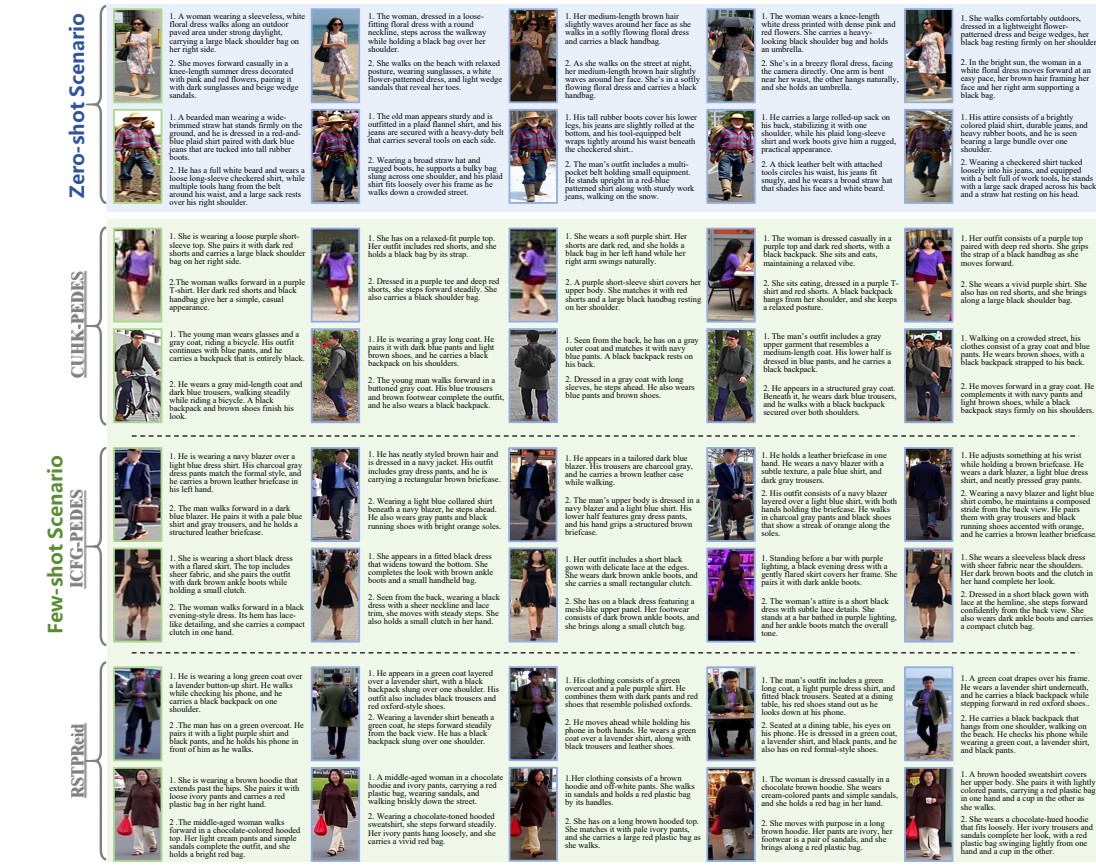

Figure A9: Representative examples of the data expanded by the TIPS framework. Each row corresponds to five images and ten image–text pairs belonging to the same identity. The leftmost image with a green border denotes the seed person image, while the four images to its right with blue borders are person images generated by the IDPG. From top to bottom, the four rows respectively illustrate: (1) typical examples of expanded data in the zero-shot setting, (2) examples of expanded data from the CUHK dataset in the few-shot setting, (3) examples from the ICFG dataset in the few-shot setting, and (4) examples from the RSTP dataset in the few-shot setting.

Table A5: Quantitative comparison of background and pose diversity between generated and real data. Background similarity is computed using Alpha-CLIP visual encoders on background masks obtained via MediaPipe segmentation; pose metrics are derived from MediaPipe Pose keypoints with center alignment and normalization.

| Data | Background Similarity ↓ | Pose Distance ↑ | Pose Similarity ↓ |
|---|---|---|---|
| CUHK (Real) | 0.915 | **0.326** | **0.701** |
| Ours (Generated) | **0.881** | 0.323 | 0.714 |

ity), where images generated from the same seed prompt diverge under different variation prompts, confirming IDPG's essential role in maintaining identity across expanded images.

For diversity assessment, we employ dedicated computational frameworks to evaluate background and pose variations. Background similarity is measured by first applying MediaPipe Lugaresi et al. (2019) for person segmentation, followed by Alpha-CLIP Sun et al. (2024) visual encoders on the isolated background regions, with cosine similarity computed between the embeddings. Pose diversity is evaluated using MediaPipe Pose to extract body keypoints, with two complementary metrics: mean Euclidean distance between normalized keypoints (pose distance) and angular similarity between flattened keypoint vectors (pose similarity).

As shown in Table A5, our generated dataset achieves greater background diversity, reflected by a substantially lower similarity score (0.881) compared with real CUHK data (0.915). Pose diversity remains highly comparable to real data, with pose distances of 0.323 (vs. 0.326) and pose similarities of 0.714 (vs. 0.701), computed over 1,000 randomly sampled identity pairs.

These quantitative findings confirm that our framework avoids repetitive background or pose patterns while maintaining realistic variation essential for robust TPR training. The MLLM filtering stage further reinforces this outcome by explicitly rejecting images lacking sufficient diversity in these dimensions.

### E.3 VISUAL DEMONSTRATION OF EXPANDED DATA

Figure A9 presents representative examples generated by the TIPS data expansion framework across different experimental scenarios. Visual inspection shows that all generated images maintain high visual fidelity without exhibiting the common artifacts frequently observed in prior pedestrian image generation approaches such as MALS Yang et al. (2023), including facial distortions, disproportionate body structures, or generally unnatural appearances. This improvement is largely attributed to the strict threshold-based filtering enforced by the MLLM.

Moreover, the five images corresponding to each identity display highly reliable identity consistency across the TIPS-generated samples. Key appearance characteristics are well preserved across all images, while visual quality remains comparable to that of real-world datasets. At the same time, these generated images exhibit rich variations in scene context, lighting conditions, poses, and clothing attributes, achieving a level of diversity that matches—if not exceeds—manually annotated TPR datasets in several aspects.

Comparisons between zero-shot and few-shot expanded datasets show that zero-shot generations tend to exhibit greater appearance diversity. This emerges because, in few-shot scenarios, the SPG becomes further aligned to the target domain distribution through training, and the LLM-generated prompts for seed person creation become more target-domain–specific. As a result, the generated images demonstrate stronger stylistic and content alignment with the target domain, leading to slightly reduced overall diversity.

This effect is also reflected in the appearance characteristics produced by the SPG. For example, in the CUHK few-shot setting, the SPG generates seed person images with a wider spectrum of resolutions, including both blurrier and sharper outputs. When the LLM-generated seed prompts do not explicitly specify environmental details, the SPG automatically produces backgrounds that are naturally consistent with dominant target-domain environments—such as academic building surroundings in CUHK and parking-lot environments in ICFG and RSTP. Remarkably, after few-shot training, the SPG even learns to reproduce domain-specific traits, such as the mosaic face occlusion characteristic of ICFG.

During the expansion stage, the IDPG maintains high identity fidelity and facial consistency while simultaneously preserving domain characteristics. It also avoids generating backgrounds or poses overly similar to those in the seed images, thereby preventing pattern repetition and ensuring natural diversity throughout the expanded dataset.

Collectively, these visual examples clearly illustrate why the TIPS-expanded data consistently yields stable and significant improvements in retrieval performance across diverse scenarios.

## DECLARATION OF THE USE OF LARGE LANGUAGE MODELS (LLMS)

The use of LLMs serves as a general-purpose assist tool throughout the research and writing process. Specifically, LLMs are used to generate diverse text prompts for the synthesis of person image datasets in the proposed Text-Image Pairs Synthesis (TIPS) framework. These models facilitate the creation of textual descriptions, ensuring high diversity and alignment with various datasets, as well as helping to filter and refine generated images. However, it is important to note that LLMs do not contribute to the ideation, structuring, or overall writing of the research paper. They are not used to assist with the formulation of the main concepts, methodology, or results discussed in this paper.

This usage complies with the guidelines set for the responsible use of LLMs, ensuring that the model's role is clearly outlined and transparent without contributing directly to the core research ideation or academic writing process.

