# OpenReview forum: "TIPS: A Text-Image Pairs Synthesis Framework for Robust Text-based Person Retrieval"
_ICLR.cc/2026/Conference — Submitted to ICLR 2026_

### Official Review · Reviewer_2BnY · 2025-10-29

**Soundness:** 3
**Presentation:** 4
**Contribution:** 4
**Rating:** 8
**Confidence:** 4

**Summary:**

The authors propose a fully automated text-image pair synthesis framework, TIPS, to address critical challenges in Text-based Person Retrieval (TPR), such as poor zero-shot adaptability and the low quality and limited practicality of existing synthetic data. For the first time, they generate a high-fidelity, identity-consistent pedestrian image dataset with controllable resolution solely from textual descriptions, and further introduce a complementary enhancement strategy—Test-Time Augmentation (TTA).

**Strengths:**

1. This paper clearly identifies the current bottlenecks in the TPR task, and the authors' motivation for proposing an automated text-image pair generation pipeline is well-justified.
2. The authors also propose a plug-and-play Test-Time Augmentation (TTA) strategy that enhances the performance of existing methods, and experimental results demonstrate the superiority of their approach.

**Weaknesses:**

1、Regarding the proposed TTA module, although the experiments demonstrate its effectiveness, the introduction of this component appears somewhat abrupt relative to the overall motivation of the paper. The TTA mechanism seems not to be conceptually aligned with the core objective of the work.
2、For the proposed dataset, the paper (including the supplementary material) does not provide detailed statistical information or descriptive analysis. This lack of dataset characterization limits the reader’s understanding of its scale, diversity, and quality.
3、The ablation study section is rather limited. For instance, one distinctive feature of TIPS is the ability to control the pixel quality of generated images. However, the paper does not investigate whether image resolution or pixel-level control affects the final retrieval performance.

**Questions:**

Detailed comments can refer to the weaknesses section.

---

> ### Author Response · Authors · 2025-11-20
>
> We sincerely appreciate your time, effort, and positive evaluation. In response to the key concerns you raised, we provide detailed clarifications below. Additionally, we have incorporated revisions addressing the major issues into the newly updated PDF, and more complete results and analyses will be included in the final version of the paper.
>
> ## **Response to Weaknesses and Questions**
>
> ### **A1 Response on the Concern That the Introduction of TTA Appears Abrupt**
>
> Thank you for pointing this out. In the current draft, the introduction of TTA may appear somewhat abrupt, and we will refine the relevant sections in the final version to improve the logical coherence. In fact, the core motivation of this work is to achieve **Robust Text-Based Person Retrieval**. To comprehensively enhance the performance of TPR methods across various scenarios, there are generally two complementary directions: advancements at the *training stage* and enhancements at the *inference stage*.
>
> From the training perspective, designing a single unified module that is universally compatible with all models and scenarios is non-trivial. For this reason, we adopt large-scale fully synthetic data expansion as a way to circumvent this challenge, and our experiments demonstrate that this strategy indeed provides consistent gains across multiple settings.
>
> From the inference perspective, we also aim to introduce a general-purpose performance enhancement mechanism. The proposed TTA module is motivated exactly by this objective. It leverages two key components: the visual encoder (trained in most TPR methods but underutilized at inference) and the SPG generator from our TIPS data pipeline, to boost test-time retrieval accuracy. Therefore, TTA is not conceptually detached from the overall motivation of the paper. Instead, it connects directly to the SPG generator and jointly serves the goal of improving the *robustness* of TPR.
>
> In the final version, we will refine the exposition to establish clearer conceptual grounding for TTA and ensure that its introduction is more natural and well-integrated into the overall narrative.
>
> ### **A2 Response on the Lack of Statistical Information and Dataset Characterization**
>
> Thank you for raising this very valuable point. The current manuscript indeed lacks comprehensive statistical descriptions and intuitive examples that characterize these datasets. In particular, we do not sufficiently present distributional statistics, diversity analysis, or visualizations that help readers better understand the scale, variety, and quality of the generated data.
>
> To address this limitation, we have added detailed dataset statistics and descriptive analyses in **Appendix E** of the revised manuscript. In addition to overall data distribution and textual attribute statistics, Appendix E now also includes **quantitative measurements of identity consistency, background and pose diversity**, as well as **representative visual examples across zero-shot and few-shot scenarios**. These additions provide readers with a clearer and more objective understanding of the dataset scale, linguistic diversity, visual variation, and the overall quality of the generated image–text pairs.

---

> > ### Author Response · Authors · 2025-11-20
> >
> > ### **A3 Response on the Limited Ablation Study and Missing Analysis of Resolution / Pixel-Level Control**
> >
> > Thank you for pointing out this important issue. The ablation studies in our initial submission are indeed not sufficiently comprehensive. Based on your suggestion, we have added several new ablation experiments in the updated version to improve the analytical depth and clarity of our method (see Appendix D.1 for full details). These additions cover:
> >
> > * the effect of image resolution control on retrieval performance,
> > * the impact of reusing zero-shot expanded data in the few-shot scenario,
> > * the influence of different LLMs, and
> > * the performance variations introduced by different MLLM filtering and captioning strategies.
> >
> > Regarding your specific concern about **whether pixel quality / image resolution control affects the final retrieval performance**, we conduct experiments using IRRA as the baseline model on CUHK under both zero-shot and few-shot settings. The results are:
> >
> > | Method | Zero-Shot R1 | Zero-Shot MAP | Few-Shot R1 | Few-Shot MAP |
> > | - | - | - | - | - |
> > | Baseline | – | – | 34.44 | 32.57 |
> > | w/o size | 51.23 | 46.85 | 54.04 | 48.79 |
> > | Low | 51.93 | 47.22 | 54.90 | 49.36 |
> > | High | 50.18 | 45.92 | 53.15 | 47.64 |
> > | Ours | **52.89** | **47.81** | **55.73** | **49.72** |
> >
> > Here:
> >
> > * **w/o size** means SPG does not receive any resolution condition during inference.
> > * **Low** means always using the lowest resolution condition seen during training.
> > * **High** means always using the highest resolution condition.
> > * **Ours** is our final design, where a **random resolution condition (sampled from those seen during training)** is used during inference.
> >
> > The results show that using random resolution conditions achieves the best performance, consistently across both zero-shot and few-shot settings. Interestingly, **always using the lowest resolution condition performs better than always using the highest resolution**. Our hypothesis is as follows: high-resolution images are easier to learn and have a more stable training distribution, potentially causing the model to overfit the clearer generative distribution. This may undermine generalization when the real test data contain more varied or lower-resolution samples. In contrast, low-resolution conditions introduce a slightly more challenging learning objective, suppressing overfitting and improving generalization. Nevertheless, using *random* resolution conditions provides the best balance between learning difficulty and generalization capability, thus achieving optimal retrieval performance.
> >
> > Beyond resolution-related experiments, we also supplement multiple additional ablations in Appendix D.1 to provide a more complete demonstration of the effectiveness and robustness of TIPS.

---

> ### Author Response · Authors · 2025-11-27
>
> Dear Reviewer 2BnY,
>
> Thank you again for your supportive review and constructive comments.
>
> As the rebuttal period is coming to an end, we wanted to take a moment to highlight that we have incorporated additional interpretability analyses into our revision. We hope these additions further strengthen the paper and solidify your positive assessment.
>
> We remain fully available to answer any final questions or provide further clarifications if needed.
>
> We sincerely appreciate your time and consideration.
>
> Best regards,
>
> Authors of Submission 20172

---

### Official Review · Reviewer_Xhbx · 2025-10-30

**Soundness:** 2
**Presentation:** 3
**Contribution:** 3
**Rating:** 4
**Confidence:** 4

**Summary:**

This paper proposes the TIPS framework, an automated system for synthesizing text-image pairs, designed to address the problem of data scarcity in text-based person retrieval tasks under zero-shot, few-shot, and cross-domain scenarios. Its core innovation lies in two diffusion-based efficient generators: SPG, which generates seed images, and IDPG, which expands images while preserving identity consistency. Additionally, it includes a comprehensive LLM/ MLLM-integrated pipeline and a test-time augmentation strategy.

**Strengths:**

1. This paper is of practical value, as it addresses the issues of identity consistency and diversity in data synthesis, thereby expanding text-image pairs for text-based person retrieval.
2. The framework is comprehensive, covering the entire process from text generation to final training data synthesis, and can be extended to other multimodal synthesis tasks.
3. The experiments on dataset quality evaluation are convincing, as demonstrated by the results under zero-shot, few-shot, and generalization scenarios presented in the paper.

**Weaknesses:**

1. The paper presents qualitative results but lacks quantitative evaluation of the identity consistency in IDPG-generated images (e.g., using a pretrained face or ReID model to compute feature similarity between generated image pairs).
2. The overall pipeline quality relies on the accuracy of the MLLM serving as a “judge.” However, the potential biases and errors of the MLLM itself may be introduced into the synthesized data, which has not been thoroughly discussed.
3. The generation cost is relatively high; although the model is lightweight, producing 400k pairs of samples still requires considerable time and computational resources.

**Questions:**

1. Could a quantitative evaluation of the identity consistency be conducted for the set of images generated by IDPG?
2. Does the TTA significantly increase the inference overhead?
3. The MLLM may make mistakes during the filtering and annotation process. Have you investigated potential errors and analyzed how these errors might affect the quality of the synthesized data and, consequently, the performance of downstream TPR models?
4. Does the generated data exhibit any “background or pose patterning” issues? Could you provide diversity statistics to illustrate this?

---

> ### Author Response · Authors · 2025-11-20
>
> Thank you very much for your valuable and insightful feedback. We carefully address each of your concerns as follows. Moreover, the newly updated PDF already includes revisions addressing the major points you raised, and the final version will further provide more comprehensive results and analyses.
>
> ## **Response to Weaknesses**:
>
> ### **A1.1 Quantitative Evaluation of Identity Consistency in IDPG-Generated Images**
>
> Thank you for raising this valuable point. Considering that faces in text-based person retrieval datasets are frequently blurred, occluded, or even invisible, face recognition models are not reliable for the direct quantitative assessment of identity consistency. Therefore, we adopt a well-trained ReID model to quantify identity consistency.
>
> Specifically, we use the SOLIDER (Swin-Base) model [2] (trained on Market1501 [1]) to extract global embeddings for each person image, and compute **cosine similarity** between image pairs as the identity consistency metric.
>
> To avoid any source overlap-induced evaluation bias, we exclude CUHK-PEDES (image sources contain Market1501), and evaluate on the **ICFG** and **RSTP** test sets:
>
> * For each identity, we compute the average similarity between two real images as the baseline.
> * We then calculate two similarities: (1) the similarity between two SPG-only generated images under the zero-shot setting (SPG Zero-Shot), and (2) the similarity between an SPG seed image and its IDPG-extended image under the zero-shot setting (SPG+IDPG Zero-Shot).
>   The results are shown below:
>
> | Data | ICFG Test Set | RSTP Test Set | SPG (Zero Shot) | SPG+IDPG (Zero Shot) |
> | - | - | - | - | - |
> | Similarity | 0.924 | 0.916 | 0.886 | 0.919 |
>
> Explanations:
>
> * The similarities for real test sets are computed by sampling one pair of real images for each identity and averaging the cosine similarities.
> * **SPG (Zero-Shot)** refers to the similarity between two images generated solely by SPG under the zero-shot configuration, using the same seed prompt combined with different variation prompts (1000 identity pairs).
> * **SPG+IDPG (Zero-Shot)** computes similarity between an SPG seed image and its corresponding IDPG-extended image (1000 identity pairs).
>
> The results indicate that:
>
> * Images generated **only by SPG** exhibit noticeably lower similarity (0.886), indicating limited ability to preserve identity.
> * After incorporating **IDPG**, the similarity rises sharply to **0.919**, which is nearly identical to (and even higher in some cases) the real-image baselines (0.924 / 0.916).
>
> The quantitative comparison provided by this experiment demonstrates that **IDPG is highly effective in maintaining identity consistency**, strongly supporting the reliability of our generation pipeline.
>
> [1] Zheng L, Shen L, Tian L, et al. Scalable person re-identification: A benchmark[C]//Proceedings of the IEEE international conference on computer vision. 2015: 1116–1124.
> [2] Chen W, Xu X, Jia J, et al. Beyond appearance: a semantic controllable self-supervised learning framework for human-centric visual tasks[C]//Proceedings of the IEEE/CVF conference on computer vision and pattern recognition. 2023: 15050–15061.

---

> > ### Author Response · Authors · 2025-11-20
> >
> > ### **A1.2 On the Potential Bias Introduced by Using an MLLM as the “Judge”**
> >
> > We agree that using an MLLM as a “judge” introduces inherent risks of bias. To tackle this issue, as illustrated in Appendix Figure A1, we take substantial measures to restrict such bias through **carefully engineered instruction design**. During the scoring process, our prompts specify all evaluation dimensions in detail and explicitly define the conditions for obtaining high scores. We also apply **strict acceptance thresholds**—for example, for 10-point scoring dimensions, *all* criteria must score ≥ 9, and binary-classification criteria must all be satisfied. These constraints significantly reduce the number of noisy samples passing filtering due to the MLLM’s subjective preferences.
> >
> > Moreover, following your suggestion, we conduct additional experiments to assess the impact of the MLLM filtering strategy on downstream performance—using IRRA as the baseline model on the CUHK dataset under both zero-shot and few-shot settings. The results are shown below:
> >
> > | No. | Method | Zero-Shot R1 | Zero-Shot MAP | Few-Shot R1 | Few-Shot MAP |
> > | - | - | - | - | - | - |
> > | 1 | Baseline | – | – | 34.44 | 32.57 |
> > | 2 | w/o Filtering | 49.45 | 45.26 | 52.13 | 47.75 |
> > | 3 | Simple Filtering | 50.73 | 46.23 | 53.54 | 48.31 |
> > | 4 | Ours | 52.89 | 47.81 | 55.73 | 49.72 |
> >
> > Here:
> >
> > * **w/o Filtering** removes the filtering stage entirely, retaining all generated samples.
> > * **Simple Filtering** drastically simplifies the filtering prompt and relies only on a binary final decision. For SPG, the simplified instruction is:
> >
> > > “Prompt: '{Seed Prompt}' Evaluate the person image considering Alignment (image–text similarity) and Fidelity (visual quality of the person image). Output only ‘yes’ or ‘no’ as the final overall result.”
> >
> > For IDPG, the simplified filtering instruction is:
> >
> > > “Evaluate Image2 for Fidelity (visual quality of the person image), Consistency (identity and outfit consistency with Image1), and Difference (whether there is a meaningful variation while preserving identity). Output only ‘yes’ or ‘no’ based on the overall assessment.”
> >
> > Because simplified filtering drastically reduces the rejection rate, we reuse as much generated data as possible in No. 2 and No. 3, and then downsample to 400K pairs per identity group to ensure a fair comparison.
> >
> > The results clearly show that **filtering is essential** for ensuring the quality of the synthesized data. Compared to having no filtering, our full filtering design improves Rank-1 by **over 3%** in both zero-shot and few-shot settings. Moreover, replacing our filtering strategy with the simplified version leads to **more than 2% performance degradation**, demonstrating that our multi-dimensional evaluation criteria and strict thresholds are also necessary to effectively constrain potential MLLM bias and prevent noisy samples from entering the final dataset.

---

> > > ### Author Response · Authors · 2025-11-20
> > >
> > > ### **A1.3 On the Concern Regarding the High Generation Cost**
> > >
> > > We understand your concern regarding computational cost. Frankly speaking, since this work represents the first systematic exploration of **fully synthetic person image–text pair construction**, and given the need to validate TIPS across zero-shot, generalization, and few-shot scenarios, training the generators and expanding data for each scenario is indeed necessary and worthwhile at this stage. And importantly, this cost is essentially **one-time** in research contexts.
> > >
> > > We will certainly release both the trained generators and the generated datasets for future research. As a result:
> > >
> > > * In **zero-shot** and **generalization** scenarios, researchers can directly reuse our generated data **without rerunning the pipeline**.
> > > * In **few-shot** scenarios, subsequent works can fine-tune on top of our released data and models, again avoiding re-training from scratch.
> > >
> > > As detailed in Appendix C.2, using two H800 GPUs:
> > >
> > > * Training **SPG** in a new scenario takes *4 hours and 19 minutes*,
> > > * Training **IDPG** takes *6 hours and 43 minutes*,
> > > * Completing the full data expansion pipeline requires approximately *135 hours*, which is far more efficient than manually annotating a dataset of equivalent scale.
> > >
> > > In practice, several strategies can greatly reduce cost. As shown in Appendix Figure A3, even generating **only one-quarter** of the full dataset already yields highly competitive retrieval performance, effectively providing a fourfold speedup. With sufficient hardware, the data generation process can also be parallelized, providing further acceleration. Training time can be significantly reduced by disabling gradient accumulation. Additionally, in few-shot scenarios, one may directly reuse the zero-shot expanded dataset without regeneration.
> > >
> > > To validate this, we report retrieval performance using IRRA as the baseline model:
> > >
> > > | Pre-training Data | CUHK-R1 | CUHK-mAP | ICFG-R1 | ICFG-mAP | RSTP-R1 | RSTP-mAP |
> > > | - | - | - | - | - | - | - |
> > > | - | 34.44 | 32.57 | 19.73 | 10.20 | 30.70 | 24.65 |
> > > | Ours (Reuse) | 53.28 | 46.98 | 40.59 | 21.73 | 51.15 | 40.03 |
> > > | Ours (Regeneration) | 55.73 | 49.72 | 45.94 | 24.75 | 54.30 | 42.00 |
> > >
> > > Here:
> > >
> > > * **Reuse** means no new data is generated; we directly reuse the zero-shot expanded dataset and fine-tune with a small amount of target-domain data.
> > > * **Regeneration** means retraining SPG and rerunning the entire expansion pipeline under the few-shot setting.
> > >
> > > The results show that simply reusing the zero-shot expanded data already provides substantial improvements, making it a cost-effective and stable option. For users aiming for peak performance, regeneration yields further gains because retraining the generators allows the synthetic distribution to better match the target domain.
> > >
> > > ## **Response to Questions:**
> > >
> > > ### **A2.1**
> > > This concern overlaps with our response provided in **A1.1**. Please refer to A1.1.
> > >
> > > ### **A2.2 On Whether TTA Significantly Increases Inference Overhead**
> > >
> > > As discussed in detail in Appendix C.2 of the paper, enabling TTA does introduce additional inference latency. However, it is important to emphasize that **the TTA module is optional**. Except for Table 4, all other experiments in the paper are conducted *without* using TTA, and therefore the core conclusions of our work are unaffected by this overhead.
> > >
> > > In practical applications, if the goal is to achieve the highest possible retrieval performance, TTA can serve as a flexible and highly compatible enhancement module that consistently improves results across a variety of TPR methods. Conversely, in scenarios where inference efficiency is critical, users can simply disable TTA and rely solely on the data expansion component, which does not introduce any additional latency.

---

> ### Author Response · Authors · 2025-11-20
>
> ### **A2.3 On Potential MLLM Errors in Filtering and Annotation, and Their Impact on Downstream TPR Performance**
>
> As discussed earlier in A1.2, we take rigorous measures in the design of filtering instructions and threshold settings to reduce potential MLLM errors during the filtering stage, and we have empirically validated the effectiveness of these mechanisms for improving TPR performance. Regarding the possible mistakes or stylistic biases introduced by the MLLM during the captioning stage, we adopt similar strategies to ensure annotation quality and mitigate negative effects on downstream retrieval.
>
> Since we utilize **Qwen2.5-VL-32B**, which has robust visual understanding and descriptive capabilities, factual errors in basic image description are rare. A more significant issue impacting TPR performance is that the MLLM may generate **sentences with overly homogeneous structures or writing styles**, whereas real TPR datasets contain captions from diverse annotators, whose linguistic habits vary greatly. If generated captions lack stylistic variety, the trained retrieval model may become less robust to natural language variations.
>
> To address this issue, we introduce as many as **120 diverse templates** covering different sentence structures and phrasing patterns during the caption generation stage, ensuring stylistic diversity. To further quantify the efficacy of this design, following your suggestion, we conduct additional controlled experiments using IRRA on the CUHK dataset under both zero-shot and few-shot settings. The results are as follows:
>
> | No. | Method | Zero-Shot R1 | Zero-Shot MAP | Zero-Shot Style Similarity | Few-Shot R1 | Few-Shot MAP | Few-Shot Style Similarity |
> | - | - | - | - | - | - | - | - |
> | 1 | Baseline | – | – | – | 34.44 | 32.57 | – |
> | 2 | w/o Template | 47.55 | 43.88 | 6.59 | 51.80 | 47.09 | 6.93 |
> | 3 | 30 templates | 50.67 | 46.31 | 5.46 | 53.35 | 47.93 | 5.59 |
> | 4 | Ours | 52.89 | 47.81 | 4.82 | 55.73 | 49.72 | 5.03 |
>
> Here:
>
> * **w/o Template** removes all template-related instructions, leaving the MLLM to freely generate captions.
> * **30 templates** reduces the template pool to 30.
> * **Ours** uses the full set of 120 templates.
>
> Since there is no universal metric tailored for stylistic similarity (as most text similarity metrics capture semantics instead of style), we use Qwen2.5-VL-32B to compute stylistic similarity scores using the following instruction, and average results over 5000 randomly sampled sentence pairs in each setting:
>
> > **Instruction:**
> > *Given two sentences, evaluate the stylistic similarity between them. Focus on factors like sentence length, complexity, use of formal or informal language, word choice, and overall flow. Do not focus on the semantic content, but rather on how similar the style and structure are between the two sentences. Provide a score between 1 and 10.*
>
> To ensure a fair comparison and avoid redundant computation, settings No. 2 and No. 3 reuse exactly the same generated images as the final version, and only regenerate the captions.
>
> The results show:
>
> * **Template-free captioning increases stylistic similarity**, leading to reduced performance due to insufficient language diversity.
> * **30 templates** provide moderate improvement.
> * **Our full 120-template design** yields the greatest stylistic diversity (lowest Similarity), and correspondingly achieves the best zero-shot and few-shot performance.
>
> These findings confirm that controlling caption style diversity is crucial, and that our template-based design effectively mitigates potential MLLM biases in annotation, yielding more robust downstream TPR models.

---

> > ### Author Response · Authors · 2025-11-20
> >
> > ### **A2.4 On Whether the Generated Data Exhibits “Background or Pose Patterning” and Quantitative Diversity Analysis**
> >
> > Regarding your concern about potential “background or pose patterning” issues, we explicitly consider this risk when designing the IDPG filtering mechanism. Specifically, we recognize that IDPG may over-reference the seed image during expansion, potentially causing the generated images to be overly similar in background, pose, or overall composition. To prevent this, the MLLM filtering stage (see Appendix A.2, *FILTER AND CAPTION*) includes a dedicated evaluation dimension that checks **whether the expanded image is excessively similar** to the seed image. Appendix Figure A2 provides typical examples that are rejected by this rule. As shown, even if the expanded image exhibits certain changes, insufficient differences in background or pose result in immediate rejection, effectively preventing patterning in the final dataset.
> >
> > To quantitatively verify the diversity of background and pose in our generated data, we design two statistical measures and compare our results with real images.
> >
> > 1. **Background Similarity**
> >    We use MediaPipe [3] to segment out the person foreground, invert the mask to obtain a background mask, and feed the image together with the background mask into the Alpha-CLIP [4] visual encoder. Alpha-CLIP focuses specifically on the masked regions and extracts background-related embeddings. We then compute cosine similarity between embeddings of two images as the background similarity metric (lower is better).
> >
> > 2. **Pose Diversity Metrics**
> >    Using MediaPipe Pose, we extract body keypoints and normalize them to eliminate scale differences. After aligning the pose centers (e.g., torso midpoint):
> >
> >    * we compute the mean Euclidean distance between corresponding keypoints as the **pose distance** (higher is better), and
> >    * flatten the keypoints into vectors and compute the angle between them as **pose similarity** (lower is better).
> >
> > We randomly select 1000 identities from the CUHK test set and compute these metrics using two real images per identity as a baseline. We then compare them against 1000 IDPG-expanded pairs generated under the zero-shot setting. Results are shown below:
> >
> > | Data | Background Similarity ↓ | Pose Distance ↑ | Pose Similarity ↓ |
> > | - | - | - | - |
> > | CUHK | 0.915 | 0.326 | 0.701 |
> > | Ours | 0.881 | 0.323 | 0.714 |
> >
> > From the table, we observe that:
> >
> > * Our generated data exhibits **even higher background diversity than real data** (i.e., lower similarity).
> > * The pose diversity of our generated images is **very close to real-world levels**.
> >
> > Therefore, **our expanded dataset does not suffer from background or pose patterning**, and in fact achieves diversity comparable to that of real data.
> >
> > [3] Sun Z, Fang Y, Wu T, et al. Alpha-clip: A clip model focusing on wherever you want[C]//Proceedings of the IEEE/CVF conference on computer vision and pattern recognition. 2024: 13019–13029.
> > [4] Lugaresi C, Tang J, Nash H, et al. Mediapipe: A framework for building perception pipelines[J]. arXiv preprint arXiv:1906.08172, 2019.

---

> ### Author Response · Authors · 2025-11-27
>
> Dear Reviewer Xhbx,
>
> Thank you again for your thoughtful review and constructive feedback.
>
> As the rebuttal period is coming to an end, we respectfully want to ensure that our previous response and the additional interpretability analyses have adequately addressed your primary concerns.
>
> We remain fully available to provide further clarification or supporting details if there are any remaining points that require elaboration.
>
> We sincerely appreciate your time and consideration.
>
> Best regards,
>
> Authors of Submission 20172

---

### Official Review · Reviewer_rPXb · 2025-10-30

**Soundness:** 3
**Presentation:** 3
**Contribution:** 3
**Rating:** 6
**Confidence:** 4

**Summary:**

This paper proposes the TIPS framework, which uses two diffusion generators (SPG/IDPG) together with an MLLM to fully automate the synthesis of high-fidelity, diverse text–pedestrian image pairs. At inference time, TTA fuses the visual features of synthesized preview images with text features, delivering steady gains without modifying the model. The method shows improvements on CUHK-PEDES, ICFG-PEDES, and RSTPReid under zero/few-shot and cross-domain settings.

**Strengths:**

1.Provides a fully automated, scalable data-generation pipeline, from prompt generation to synthesis, data filtering, and automatic description, capable of batch-producing high-quality text–pedestrian image pairs.

2.Achieves significant gains in zero/few-shot settings with strong sample efficiency.

3.Requires no network modifications: at inference, fusing “preview image” features with text features enhances consistency and boosts performance.

**Weaknesses:**

1.The pipeline is relatively complex overall, relying on LLMs/MLLMs and generators, which raises implementation complexity.

2.Each scenario requires training the generators first; expanding to 400k pairs incurs substantial computation and time costs.

3.A preview image must be generated at inference, adding 2.75s per query; latency increases markedly for methods without reranking.

4.SPG may produce appearance/identity inconsistencies across runs under the same prompt, so it depends on IDPG and MLLM filtering, failures in these stages can degrade quality.

5.The TTA fusion coefficient α requires empirical tuning and may need to be adjusted across methods/datasets.

6.Data scoring and description are produced by an MLLM, so stylistic biases or preferences may be injected into the synthetic data, affecting downstream distributions.

**Questions:**

1.If identity drift occurs, what is its frequency, and what proportion of cases are corrected by the IDPG + MLLM filtering?

2.Is the per-scenario training time cost excessively high?

3.Please quantify the stylistic diversity and similarity of the generated texts, compare results across different LLMs/templates, and evaluate whether MLLM-produced descriptions introduce stylistic bias that affects downstream retrieval.

4.In scenarios with available annotations, SPG can be trained directly on target-domain image–text pairs. Please explain how you prevent leakage or overlap with the test distribution/identities.

If these concerns are addressed, I will raise my score.

---

> ### Author Response · Authors · 2025-11-20
>
> We appreciate your efforts and time, as well as your overall positive assessment. In response to your critical concerns, detailed clarifications are provided below. In addition, we have reflected the key revisions in the updated PDF, and the final paper will present an even more complete set of results and analytical discussions.
>
> ## **Response to Weaknesses:**
>
> ### **A1.1 Regarding the Complexity of the Pipeline**
>
> We acknowledge your concern that the overall pipeline appears complex. However, each component in the pipeline is essential for achieving our objective of generating **fully synthetic and identity-consistent person image–text pairs**.
>
> First, no existing generator can directly produce multiple identity-consistent person images from text. Therefore, training both the **Seed Person Image Generator (SPG)** and the **Identity Preservation Generator (IDPG)** is necessary, as these two components are responsible for learning the core ability to generate consistent-identity images.
>
> Second, the diversity of generated person images depends heavily on the expressive richness of the prompts. Using only a small number of handcrafted templates or limited random combinations cannot yield sufficient variation. Hence, we employ LLMs to **automatically generate high-quality and semantically diverse prompts**, which is a crucial step for expanding the variability of appearance, scenes, and attributes.
>
> After image generation, diffusion-based models inevitably introduce noise or occasional artifacts. To ensure the reliability of the final dataset, we leverage **MLLMs for filtering**, assessing identity consistency, visual fidelity, and the absence of undesirable artifacts. This step is pivotal for maintaining high-quality outputs.
>
> Finally, to guarantee both textual diversity and accurate text–image alignment, we use **MLLMs to regenerate captions** for each synthesized image. This enhances descriptive richness while preserving alignment quality.
>
> In summary, TIPS achieves the automatic generation of fully synthetic, identity-consistent, high-quality image–text pairs for the first time through four organically connected components. It avoids privacy concerns while striking a balance between effectiveness and complexity.
>
> ### **A1.2 Regarding the Training and Computational Cost of Expanding to 400K Image–Text Pairs**
>
> Frankly speaking, since this work represents the first systematic exploration of **fully synthetic person image–text pair construction**, and given the need to validate TIPS across zero-shot, generalization, and few-shot scenarios, training the generators and expanding data for each scenario is indeed necessary and worthwhile at this stage. And importantly, this cost is essentially **one-time** in research contexts.
>
> We will certainly release both the trained generators and the generated datasets for future research. As a result:
>
> * In **zero-shot** and **generalization** scenarios, researchers can directly reuse our generated data **without rerunning the pipeline**.
> * In **few-shot** scenarios, subsequent works can fine-tune on top of our released data and models, again avoiding re-training from scratch.
>
> As detailed in Appendix C.2, using two H800 GPUs:
>
> * Training **SPG** in a new scenario takes *4 hours and 19 minutes*,
> * Training **IDPG** takes *6 hours and 43 minutes*,
> * Completing the full data expansion pipeline requires approximately *135 hours*, which is far more efficient than manually annotating a dataset of equivalent scale.
>
>
> ### **A1.3 The Additional Inference Latency Introduced by Preview Image Generation**
>
> It is important to emphasize that **the TTA module is optional**. Except for Table 4, all other experiments in the paper are conducted *without* using TTA, and therefore the core conclusions of our work are unaffected by this overhead.
>
> In practical applications, if the goal is to achieve the highest possible retrieval performance, TTA can serve as a flexible and highly compatible enhancement module that consistently improves results across a variety of TPR methods. Conversely, in scenarios where inference efficiency is critical, users can simply disable TTA and rely solely on the data expansion component, which does not introduce any additional latency.

---

> > ### Author Response · Authors · 2025-11-20
> >
> > ### **A1.4 Response on Appearance/Identity Consistency and the Reliability of the Filtering Mechanism**
> >
> > It should be noted that, as described in the manuscript, all images—whether generated by SPG or by IDPG—must pass through a **strict MLLM-based filtering process** before being retained in the final expanded dataset. Only samples that simultaneously meet the requirements for identity consistency, visual fidelity, and text–image alignment are preserved. This design strictly prevents inconsistencies or noise introduced during the generation from propagating into the final training set.
> >
> >
> > ### **A1.5 Response on the TTA Fusion Coefficient α**
> >
> > Indeed, the optimal α value may vary slightly with different methods or datasets. However, according to our default configuration, setting α = 0.6 consistently yields substantial improvements across all three datasets in Table 4. Furthermore, Appendix D.1 provides performance curves for different α values across multiple methods, from which it is clear that **as long as α is conservatively chosen within the range (e.g., 0.4–0.8), the performance achieves stable and substantial gains**, indicating that the method is not overly sensitive to the selection of α.
> >
> > More importantly, as discussed in Appendix D.1, if one wishes to pursue the best possible performance in practical applications, TTA can be seamlessly integrated into the inference pipeline of any TPR method without modifying training. Therefore, the optimal α can be efficiently determined by performing a grid search on the validation set purely during inference. This makes the tuning of α both low-cost and highly practical.
> >
> > ### **A1.6 Response on Potential Stylistic Bias Introduced by MLLMs**
> >
> > It is true that MLLMs may introduce potential stylistic bias. To mitigate this issue, we carefully design instruction prompts (as shown in Figure A1). During scoring, we explicitly specify in the MLLM instructions the evaluation dimensions, the criteria for obtaining higher scores, and strict thresholds. For example, for dimensions scored on a 1–10 scale, we require each score to be at least 9, and for binary criteria, all conditions must be satisfied. These stringent rules effectively suppress the impact of any single-dimension bias and prevent noisy samples from passing the filter.
> >
> > Moreover, when generating captions for person images, we introduce **120 templates** covering diverse sentence structures and stylistic patterns. These templates ensure rich linguistic variation while guiding the MLLM toward a predetermined writing style. (See Reply A2.3 for further details.)
> >
> > ## **Response to Questions:**
> >
> > ### **A2.1 Frequency of Identity Drift and the Corrective Ability of IDPG + MLLM Filtering**
> >
> > In Appendix A.2 (“FILTER AND CAPTION”) of the original paper, we report the overall MLLM rejection rates for SPG across different scenarios (19.6% in the zero-shot setting, 17.5% in the few-shot setting, and 14.6% in the generalization setting), as well as the overall rejection rates for IDPG-expanded images (which remain relatively stable at an average of 36.2%).
> > Here, in response to your concern, we supplement the finer-grained dimension-wise rejection rate statistics as follows.
> >
> > For the **SPG** outputs:
> >
> > * **Zero-shot setting**: 15.5% low-fidelity person images, 4.6% text–image misalignment, and 0.5% where both issues occur simultaneously.
> > * **Few-shot setting**: 14.3% low-fidelity, 3.6% misalignment, and 0.4% both.
> > * **Generalization setting**: 11.9% low-fidelity, 3.1% misalignment, and 0.4% both.
> >
> > For the **IDPG** outputs, which remain relatively stable across all scenarios, the rejection rates broken down by dimension are as follows:
> >
> > * **Low-fidelity person images**: 23.0%
> > * **Identity inconsistency**: 10.5%
> > * **Insufficient difference from the seed image**: 3.9%
> > * **Total rejection rate**: 36.2%
> >
> > The “identity drift” you are concerned about directly corresponds to the **identity inconsistency** category, which occurs at approximately **10.5%** before filtering. Given that our MLLM evaluation employs highly stringent thresholds, manual review of the *retained* data reveals no identity-inconsistent cases, indicating that the *actual* identity drift rate in the final dataset is **below 10.5%**.
> >
> > In the updated PDF, we additionally provide Appendix Figure A8 showing examples with an identity-consistency score of 8. These image pairs are visually almost indistinguishable in identity, yet our threshold requires a score ≥ 9. This further demonstrates that our filtering pipeline effectively corrects identity drift and ensures the identity reliability of the final expanded dataset.

---

> > > ### Author Response · Authors · 2025-11-20
> > >
> > > ### **A2.2**
> > > This concern overlaps with our response provided in **A1.2**. Please refer to A1.2.
> > >
> > > ### **A2.3 Quantifying Stylistic Diversity and Evaluating LLM/Template Influence on Retrieval Performance**
> > >
> > > To evaluate how these stylistic choices affect downstream retrieval performance, we conduct ablation experiments by comparing IRRA on the CUHK dataset under both zero-shot and few-shot settings.
> > >
> > > | No. | Method | Zero-Shot R1 | Zero-Shot MAP | Zero-Shot Style Similarity | Few-Shot R1 | Few-Shot MAP | Few-Shot Style Similarity |
> > > | - | - | - | - | - | - | - | - |
> > > | 1 | Baseline | – | – | – | 34.44 | 32.57 | – |
> > > | 2 | w/o Filtering | 49.45 | 45.26 | 4.75 | 52.13 | 47.75 | 5.01 |
> > > | 3 | w/o Template | 47.55 | 43.88 | 6.59 | 51.80 | 47.09 | 6.93 |
> > > | 4 | 30 templates | 50.67 | 46.31 | 5.46 | 53.35 | 47.93 | 5.59 |
> > > | 5 | Ours | 52.89 | 47.81 | 4.82 | 55.73 | 49.72 | 5.03 |
> > >
> > > Where:
> > >
> > > * **w/o Filtering** removes all MLLM-based filtering; generated images are kept unchanged.
> > > * **w/o Template** removes the template-related components from the MLLM captioning instruction.
> > > * **30 templates** uses a reduced template set (size = 30).
> > > * **Ours** corresponds to the full TIPS captioning strategy with 120 templates and full filtering.
> > >
> > > Existing metrics focus on semantics rather than style, so we evaluate stylistic similarity using Qwen2.5-VL-32B as the evaluator with the following instruction:
> > >
> > > > *Given two sentences, evaluate the stylistic similarity between them. Focus on factors such as sentence length, complexity, formality, word choice, and overall flow. Do not evaluate semantic content. Output a score from 1 (no stylistic similarity) to 10 (high stylistic similarity). The two sentences are: \[Sentence 1] and \[Sentence 2].*
> > >
> > > We randomly sample 5000 sentence pairs for each setting and compute the mean stylistic similarity.
> > >
> > > To ensure fairness and avoid unnecessary computation, we reuse previously generated image–text data wherever applicable:
> > >
> > > * In **w/o Filtering**, we keep all generated images and uniformly subsample to 400K pairs.
> > > * In **w/o Template** and **30 templates**, we reuse the same filtered images as in **Ours**, but regenerate captions using the modified template sets.
> > >
> > > The results show that filtering is essential for improving image quality and brings over **3% Rank-1 improvement** in both zero-shot and few-shot settings. Furthermore, template-based caption diversification plays a crucial role: removing templates leads to worse performance even compared with unfiltered data. Increasing template diversity improves robustness to varied textual expressions, thereby enhancing retrieval performance.
> > >
> > > In addition, following your suggestions, we add an additional experiment focusing on LLMs to examine how different model scales influence downstream retrieval performance. Specifically, we build upon IRRA as the base model and conduct comparisons on the CUHK dataset under both zero-shot and few-shot settings.
> > >
> > > The results of *LLM scales* are shown below:
> > >
> > > | LLM | Zero-Shot R1 | Zero-Shot mAP | Few-Shot R1 | Few-Shot mAP |
> > > | - | - | - | - | - |
> > > | Baseline | – | – | 34.44 | 32.57 |
> > > | Qwen3-8B | 51.95 | 47.29 | 54.99 | 48.32 |
> > > | Qwen3-14B | 52.53 | 47.76 | 55.60 | 49.18 |
> > > | Ours (Qwen3-32B) | 52.89 | 47.81 | 55.73 | 49.72 |
> > >
> > > From these results, we observe that replacing our default LLM with smaller or weaker models leads to only marginal performance degradation. This indicates that the LLM’s primary role within our framework is simply to generate **semantically reasonable and stylistically diverse prompts**. The diversity itself is largely driven by our template-based and randomized design (illustrated in Appendix Figure A1), rather than the intrinsic capability of the LLM. Additionally, the instruction prompts include high-quality exemplars, which ensures that even a smaller LLM produces well-formed outputs suitable for image generation.
> > >
> > > Moreover, since the entire pipeline subsequently applies a strict MLLM-based filtering stage to remove low-quality samples, the influence of LLM limitations is further mitigated.
> > >
> > > Overall, these experiments demonstrate that our framework exhibits **strong robustness to LLM variation**. The method does not rely heavily on the scale or raw generation capability of the LLM, and stylistic bias introduced at this stage has minimal impact due to the combined effects of template-driven diversification and the subsequent MLLM quality control.

---

> ### Author Response · Authors · 2025-11-20
>
> ### **A2.4 Preventing Leakage When Using Target-Domain Annotations in the Few-Shot Setting**
>
> Sorry for causing a minor misunderstanding. In our experimental setup, the “target domain” refers specifically to the **training split** of a given dataset, not the test split. Within the same dataset, the training and test sets share the same image source and annotation style, and therefore belong to the same distribution; however, they are strictly non-overlapping, consistent with the standard protocol in person retrieval benchmarks.
>
> For example, in the few-shot setting on ICFG, we randomly sample **1% of the identities from the ICFG training set** as the available target-domain data, as described in Lines 410–413 of the main text. Because the dataset ensures that the training and test identities do not overlap, using these few training samples for SPG training **does not introduce identity leakage or distribution contamination**.
>
> It is also worth noting that this 1% subset corresponds to only **31 identities** in ICFG, a scale that is easy to annotate manually in real-world applications. This further demonstrates the practical value of our method under few-shot conditions: with only a very small number of real annotated samples, significant performance improvements can be achieved, while strictly avoiding any leakage of test identities.

---

> ### Author Response · Authors · 2025-11-27
>
> Dear Reviewer rPXb,
>
> Thank you again for your supportive review and constructive comments.
>
> As the rebuttal period is coming to an end, we wanted to take a moment to highlight that we have incorporated additional interpretability analyses into our revision. We hope these additions further strengthen the paper and solidify your positive assessment.
>
> We remain fully available to answer any final questions or provide further clarifications if needed.
>
> We sincerely appreciate your time and consideration.
>
> Best regards,
>
> Authors of Submission 20172

---

### Official Review · Reviewer_Ao47 · 2025-10-30

**Soundness:** 2
**Presentation:** 1
**Contribution:** 2
**Rating:** 2
**Confidence:** 4

**Summary:**

This paper proposes a Text-Image Pairs Synthesis (TIPS) framework to address practical challenges of TPR in real-world scenarios, including zero-shot adaptation, few-shot adaptation, and robustness issues. Two person image generators, SPG and IDPG, are introduced to synthesize realistic, identity-consistent pedestrian images. Additionally, TIPS incorporates a caption generator and a filtering mechanism to enhance data quality. Furthermore, a test-time adaptation (TTA) method is proposed to further improve retrieval accuracy.

**Strengths:**

1. The paper provides a comprehensive exploration of practical challenges in TPR tasks, such as zero-shot adaptation, few-shot adaptation, and robustness, which are critical for real-world applications.
2. The experiments are extensive and the analysis is in-depth, providing valuable empirical insights.

**Weaknesses:**

1. **Logical Inconsistency**: In the Introduction, the paper argues that existing methods typically rely on real person images, limiting extensibility and scenario diversity. However, in the methodology, the collection of training data in this work also involves gathering real-person images, which appears inconsistent with the stated motivation.
2. **Presentation and Reproducibility**: The descriptions of SPG and IDPG in the methodology section are rather opaque, making it difficult to fully understand the specific generation processes. Moreover, the correspondence between the textual descriptions and Figure 2 is unclear, which further hinders comprehension. Additionally, the writing in the methods section lacks technical rigor and professionalism. For example, in the S3 stage, the paper merely states that the outputs are "further evaluated for identity and outfit consistency with the seed image," but does not specify how MLLMs are utilized for evaluation, what the evaluation criteria are, or how generation quality and identity consistency are measured. Such methodological details are essential for reproducibility and for ensuring the technical soundness of the proposed approach.
3. **Novelty**: The proposed framework is largely an engineering integration of existing generation techniques for data augmentation under different scenarios. While practically valuable, the paper lacks substantial methodological innovation, which may limit its impact on future research.
4. **Experimental Setup**: The zero-shot setting samples images from CUHK03, CUHK02, Market-1501, MSMT17, and VIPER. However, the downstream dataset CUHK-PEDES contains images from CUHK03, Market-1501, and VIPER, while ICFG-PEDES and RSTPReid contain images from MSMT17. This setup may lead to identity overlap, which contradicts the claimed zero-shot setting. Additionally, there is a concern that test images from these datasets may inadvertently be included in the training set, potentially leading to information leakage.

**Questions:**

1. In Table 3 (generalization scenario), what is the difference between "raw" and "ours" in the training data? Please clarify this in the paper to avoid confusion.

2. The Introduction states that existing datasets suffer from poor text-image alignment, yet recent works [1,2] have focused on person image captioning. How does the proposed caption generation method ensure higher quality and greater diversity compared to these methods?

   [1] Jiang J, Ding C, Tan W, et al. Modeling Thousands of Human Annotators for Generalizable Text-to-Image Person Re-identification[C]//Proceedings of the Computer Vision and Pattern Recognition Conference. 2025: 9220-9230.

   [2] Tan W, Ding C, Jiang J, et al. Harnessing the power of mllms for transferable text-to-image person reid[C]//Proceedings of the IEEE/CVF Conference on Computer Vision and Pattern Recognition. 2024: 17127-17137.

**Details Of Ethics Concerns:**

No Ethics Concerns

---

> ### Author Response · Authors · 2025-11-20
>
> Thank you very much for the time and effort you invested in evaluating our work. We have addressed each of your concerns carefully as follows. Moreover, we have incorporated the key revisions into the newly updated PDF, and the final version will be further improved.
>
> ## **Response to Weaknesses**:
>
> ### **A1.1 Regarding the Issues of Logical Inconsistency in Motivation**
>
> There is likely a misunderstanding here:
> 1) The existing methods referenced in our paper (such as LUPerson-T and LUPerson-M) are considered to “**rely on real person images**”, because they indeed use collected real person images to construct datasets, and most of their technical effort is devoted to adding textual descriptions for these real person images. Obviously, the scalability of these approaches is limited by the amount of real data available, and privacy risks inevitably arise from direct use of identifiable human images.
>
> 2) Our method is fundamentally different. Under the zero-shot setting, we only require 300 real person images that do not overlap with the target domain to train the Seed Person Image Generator (SPG) and the Identity Preservation Generator (IDPG). Following this training, we can synthesize a large number of entirely new person images, and importantly, **the final expanded dataset contains no real person images—the dataset consists entirely of generated images**. This enables unlimited data expansion in principle while significantly reducing privacy risks, thereby overcoming the strong dependence on real images observed in existing works.
>
> 3) Moreover, it is important to clarify that our paper does not claim to completely eliminate real data throughout the entire pipeline. Instead, we highlight that our final dataset is fully synthetic and that our approach significantly mitigates privacy risks. The motivation of our work remains consistently focused on addressing the challenges of zero-shot adaptation, few-shot adaptation, and robustness in text-based person retrieval (TPR), rather than attempting to completely remove real data from the process.
>
> 4) Additionally, our SPG can, in fact, be trained **without relying on any person-containing images**. The real person images we collect primarily serve to facilitate the model’s learning of scene, viewpoint, and other image-style characteristics, and to regulate image clarity under fixed-resolution inputs. Since LoRA finetuning does not impair the text–image alignment capabilities of the base model, the final image content is still predominantly determined by the input text prompt.
>
> To further verify this, we conduct an additional experiment. We preserve the size distribution of the original 300 real person images but replace them with 300 randomly cropped environmental images from the PRW dataset [1] to train SPG. For data expansion, we reuse all prompts from the zero-shot setting in the manuscript to ensure a fair comparison. We then train IRRA using the same settings as in the main text. The results are shown below.
>
> | Pre-training Data | Scale | CUHK-R1 | CUHK-mAP | ICFG-R1 | ICFG-mAP | RSTP-R1 | RSTP-mAP |
> | - | - | - | - | - | - | - | - |
> | MALS | 1.5M | 19.21 | 18.72 | 7.88 | 3.49 | 22.50 | 16.94 |
> | LUPerson-T | 400K | 20.06 | 19.24 | 10.46 | 4.11 | 22.10 | 16.79 |
> | LUPerson-M | 400K | 48.07 | 44.12 | 27.35 | 13.95 | 42.95 | 32.46 |
> | Ours (PRW) | 400K | 51.75 | 47.04 | 33.11 | 16.89 | 47.90 | 38.64 |
> | Ours (Raw) | 400K | 52.89 | 47.81 | 33.16 | 17.15 | 48.65 | 39.04 |
>
> * **Ours (Raw)**: SPG trained using 300 real person images (original setting in the manuscript)
> * **Ours (PRW)**: SPG trained using 300 environmental images with matched size distribution
>
> The results show that even when SPG is trained without any real person images, our method still achieves strong performance improvements across all metrics. This confirms that the synthetic data generated by TIPS remains highly effective even when SPG training is fully independent of real person imagery, further supporting the motivation of our work.
>
> [1] Zheng L, Zhang H, Sun S, et al. Person re-identification in the wild[C]//Proceedings of the IEEE conference on computer vision and pattern recognition. 2017: 1367-1376.

---

> > ### Comment · Reviewer_Ao47 · 2025-11-26
> > **Use of Real Person Image**
> >
> > Thank you for the detailed response. However, I remain concerned about the clarity and rigor of the manuscript’s claims. In line 39, the paper states that existing methods “are usually based on real person image, limiting their extensibility and scenario diversity,” which strongly suggests that your method does not rely on real person images. Yet, both the manuscript and your rebuttal indicate that your approach still requires 300 real person images to train the SPG and IDPG. This creates confusion about whether your method truly overcomes the limitations you highlight. Moreover, it is unclear how using only 300 real person images can guarantee sufficient diversity in the generated dataset.

---

> ### Author Response · Authors · 2025-11-20
>
> ### **A1.2 Regarding Presentation and Reproducibility**
>
> Regarding the presentation issue, we respectfully believe your score of 1 may not fully reflect the overall quality of our presentation, given that the other three reviewers award an average score of 3.33 for this aspect. However, if the reviewer can point out the specific sections or paragraphs that you consider “opaque”, we will certainly make targeted improvements to them in the revised version.
>
> 1) For the description of the Seed Person Image Generator (SPG), Figure 2 (Stage 1) logically illustrates its core process: during image–text pair training, the width and height of the original image are extracted as conditional inputs and concatenated with the textual representations, while the resized person image is used as the supervision target. The textual description in the main text is consistent with this figure.
>
> 2) For the Identity Preservation Generator (IDPG), Figure 2 (Stage 1) explicitly shows how the reference image is compressed via a VAE encoder into latent features, which are then concatenated with noise along the spatial dimension (forming a **“diptych-like”** structure). The model learns to generate the target image under the guidance of relative prompts, with supervision applied only to the right half of the diptych. The inference procedure exactly mirrors this training mechanism. The main text follows the same logic as the figure.
>
> 3) Regarding reproducibility, due to space constraints, the main text focuses on presenting the performance across all evaluation settings while prioritizing key information. However, as stated in the REPRODUCIBILITY STATEMENT, the Appendix provides comprehensive technical details, including:
>
> * the full instruction templates used for all LLMs and MLLMs (Appendix A.2, Figure A1),
> * the specific training configurations for SPG and IDPG,
> * a step-by-step explanation of the data expansion pipeline,
> * detailed filtering criteria, evaluation dimensions, and scoring thresholds used by MLLMs in Stage S3,
> * representative examples of filtered-out data (Figure A7).
>
> In particular, the concern about Stage S3—how MLLMs perform evaluation, the criteria used, and the methodology for measuring identity and outfit consistency—has been fully documented in the *FILTER AND CAPTION* subsection of Appendix A.2 (bottom of page 16 and top of page 17). This includes the multi-dimensional scoring rules, the 1–10 scoring requirements, and the binary decision conditions for determining whether a sample exhibits sufficient variation.
>
> Furthermore, all code will certainly be made publicly available to ensure the reproducibility of the proposed method.
>
> ### **A1.3 Concern on Novelty**
>
> It should be noted that although we are not the first to leverage generated images to enhance person retrieval performance, our synthetic approach is fundamentally different from existing ones and outperforms SOTA methods with significant advantages. This is attributed to the fact that both the overall design of the TIPS framework and its two key generation components—the Seed Person Image Generator (SPG) and the Identity Preservation Generator (IDPG)—embody distinct and substantive originality:
>
> First, TIPS itself represents a new paradigm. To the best of our knowledge, this is the first framework in the person retrieval field that enables **automated construction of large-scale, fully synthetic training datasets without using any real data in the final dataset**. No prior work provides a comparable pipeline, and this novelty is positively recognized by the other three reviewers.
>
> Second, at the component level, our approach surpasses mere engineering integration and introduces new mechanisms in the generation process:
>
> * **SPG:** We incorporate a *resolution-conditioning mechanism* that allows the model to generate pedestrian images with **controllable clarity at a fixed resolution**, even when trained on an extremely limited amount of data. To our knowledge, no prior work in person-centric generation adopts a comparable functional capability or architectural design.
>
> * **IDPG:** We propose a *diptych-style latent-space concatenation* structure, enabling the model to generate multiple identity-consistent person images under the guidance of relative prompts. Existing identity-consistent generation methods focus primarily on facial consistency, and they neither generalize to full-body pedestrian images nor employ a similar architectural idea.
>
> For these reasons, we believe that TIPS offers contributions that far extend mere beyond engineering integration; it provides **novel methodological insights at both the framework and component levels**. We will further strengthen the presentation of these innovations in the final version.

---

> > ### Comment · Reviewer_Ao47 · 2025-11-26
> > **Presentation and Novelty**
> >
> > **Methodological Clarity**: (1) The manuscript mentions the "resolution condition embedding" $C_{size}$ size in line 214, but it is not clear how this embedding is incorporated into Equation (1). (2) The terms “original condition $C$” and “original image $Q$” are not clearly defined, and their use is not sufficiently professional. The manuscript does not explain what these terms specifically refer to or their roles in the method. I suggest the authors provide precise definitions and clarify how these elements are used in the overall framework to improve the clarity and professionalism of the presentation.
> >
> > **Novelty**: Regarding the SPG, the approach of controlling generated image resolution via a resolution condition is not clearly positioned. It is unclear whether similar techniques have already been explored in previous works (not limited to the TPR field), or how your method compares to existing multi-resolution image generation approaches. The manuscript does not discuss related work in this area, nor does it highlight the differences or advantages of your method. I recommend that the authors explicitly discuss the relationship between their approach and prior work on multi-resolution image generation, making it difficult to assess the originality and innovation of the proposed approach.

---

> ### Author Response · Authors · 2025-11-20
>
> ### **A1.4 Concerns on Data Leakage and the Zero-Shot Setting**
>
> We fully agree with your emphasis on the rigor of the zero-shot setting. In fact, during the experimental design phase, we have carefully considered the potential identity overlap and information leakage, and implement explicit measures to avoid these issues.
>
> As stated in Lines 333–336 of the manuscript—and reiterated in Line 767 of the Appendix—we ensure that the sampled IDs used for SPG/IDPG training do not overlap with any identities appearing in the downstream TPR datasets. The exact statement is:
>
> > “For the zero-shot setting, we uniformly select 100 IDs from five datasets (CUHK03, CUHK02, Market-1501, MSMT17, and VIPER), ensuring that these IDs do not overlap with the image sources in the TPR artificial dataset.”
>
> This means that the 100 IDs sampled from the five datasets are **strictly non-overlapping** with the image sources of the datasets used for TPR evaluation. Therefore, neither identity overlap nor information leakage can occur under our zero-shot configuration.
>
> To further ensure reproducibility and transparency, we will publicly release the complete list of sampled IDs along with their dataset origins when open-sourcing our code and data. This will ensure that others can easily verify that the zero-shot constraints are strictly satisfied.
>
> Moreover, as demonstrated in the supplementary experiment in Response **A1.1**, even when SPG is trained **without any real person images at all** (using only environmental images), our method still yields significant improvements in retrieval performance. This further confirms that:
>
> * our reliance on real data is minimal,
> * performance gains are not the result of any form of leakage, and
> * the entire training pipeline is robust and compliant with the zero-shot assumption.
>
> ## **Response to Questions:**
>
> ### **A2.1 Clarification on the Distinction Between “Raw” and “Ours” in Table 3**
>
> Thank you for pointing out the potential confusion. In this table:
>
> * **Raw:** refers to the performance obtained by directly training IRRA on the source dataset, followed by evaluation on the target dataset without any data augmentation.
> * **Ours:** refers to the performance achieved by training IRRA on the combined data of the source dataset and the TIPS-generated expanded dataset, followed by evaluation on the target dataset.
>
> We have added an explicit explanation in the revised version to avoid potential misunderstanding among readers.

---

> > ### Author Response · Authors · 2025-11-20
> >
> > ### **A2.2 Regarding the Caption Quality and Diversity**
> >
> > It is important to clarify that the study focus of our work is fundamentally different from that of the two papers you provided. Our TIPS framework aims to **automatically synthesize image–text pairs**, with our core innovation being the ability to **generate multiple identity-consistent person images purely from text**. In contrast, the two cited works focus on generating captions for **existing real person images** using MLLMs. Their task is to *describe real images*, whereas ours is to *construct new identities and generate corresponding images from scratch*. These are inherently different problem settings.
> >
> > Even so, we carefully examine the issue of caption quality. Both cited works explicitly note in their limitations that using MLLMs for annotation inevitably introduces **hallucination**, which leads to suboptimal text–image alignment, and no effective mitigation strategy has been proposed to date.
> >
> > As recent studies [2] demonstrate, **larger models generally exhibit stronger alignment capability, better interpretability, and lower hallucination rates**. The two cited methods rely on Qwen-VL-7B as the image–text alignment model, while TIPS employs the substantially larger **Qwen2.5-VL-32B**, a difference that should theoretically reduce hallucinations and improve text–image alignment. Although caption alignment is not the primary focus of our work, this difference remains meaningful and potentially beneficial.
> >
> > Regarding diversity, the distinction between the approaches is even more fundamental:
> >
> > * Captioning methods based on real person images are naturally constrained by the content and variation present in the original images.
> > * In contrast, our approach does not rely on real images and can flexibly manipulate generation prompts to control a wide range of attributes, such as **scene, age, resolution, clothing, orientation, weather, action, lighting, and obstruction** (as shown in Figure 4 of the manuscript).
> >
> > Real datasets are inherently limited by real-world capture conditions, making it difficult to achieve such comprehensive and fine-grained diversity. Our TIPS generation pipeline possesses a clear advantage in this regard.
> >
> > [2] Shen S, Qi Z, Sun J, et al. Enhancing Pre-trained Representation Classifiability can Boost its Interpretability[C]//The Thirteenth International Conference on Learning Representations. 2025.

---

> > ### Comment · Reviewer_Ao47 · 2025-11-26
> > **Zero-shot Setting**
> >
> > Thank you for your detailed explanation regarding the zero-shot setting and data leakage prevention. However, I would like to note that for text-based person retrieval (TPR), the use of non-overlapping identities between training and testing is already the standard practice, and TPR is inherently a zero-shot task. Overstating the zero-shot aspect in the manuscript may give readers unfamiliar with TPR the incorrect impression that this is a novel or unique setting for your work.

---

> ### Author Response · Authors · 2025-11-26
>
> We appreciate the reviewer for highlighting the specific phrasing that might lead to ambiguity. We have revised the corresponding sentence (**Line 39 and Line 128**) to prevent any potential misunderstanding.
>
> Our intended meaning, consistent with our previous explanation in **Response A1.1**, is that related methods typically incorporate real data directly into their **final training image-text pairs**. The revised text now clarifies this distinction. We have NEVER claimed to completely eliminate real data from the entire pipeline. Instead, we emphasize that our **final expanded dataset is fully synthetic** and that our method significantly mitigates privacy risks. Our motivation from start to finish remains focused on addressing the challenges of zero-shot adaptation, few-shot adaptation, and robustness in TPR, where we have achieved significant performance improvements to support our main contributions.
>
> Regarding the concern about diversity with only 300 images:
> Unlike other methods that merge real images into their final datasets (where 300 images would indeed be insufficient to ensure diversity), we use 300 images solely to train the **generative capabilities** of the model—specifically, the surveillance-style generation for SPG and the identity-consistency capability for IDPG. To our knowledge, no previous methods have achieved these capabilities in TPR.
>
> As explicitly detailed in **Response A1.1 (Point 4)**, since LoRA fine-tuning does not impair the text-image alignment capabilities of the pre-trained base model, the content of the generated images remains predominantly determined by the input text prompts. Consequently, the visual diversity is driven by the textual diversity. As shown in **Appendix A.2** and **Figure A1**, our carefully designed LLM-driven instruction generation strategy ensures substantial textual diversity, which effectively enhances the diversity of the generated dataset. We have added a discussion on this mechanism to **Lines 272-276** of the revised manuscript to facilitate reader understanding. Furthermore, the newly added **Appendix E** provides both objective statistics and visual examples to empirically demonstrate the high diversity of the expanded data.

---

> ### Author Response · Authors · 2025-11-26
>
> **1. Methodological Clarity**
>
> We sincerely appreciate the reviewer for their meticulous reading and for pointing out the ambiguities in our mathematical formulation.
>
> Following your valuable suggestions, we have **substantially restructured and rewritten Section 3.1**. The specific improvements are as follows:
>
> 1). **Explicit Definition of Encoder Reuse:** We have clarified that $\tau$ represents the **pre-trained text encoder**. To address the confusion regarding the "resolution condition," we now explicitly describe how resolution parameters are treated as textual tokens ($P_{\text{size}}$) and processed by $\tau$. This highlights that we reuse the pre-trained semantic space rather than initializing a new module.
>
> 2). **Rigorous Mathematical Formulation:** We have replaced the colloquial terms (e.g., "original condition", "original image") with formal mathematical notations. Specifically:
> * $Z_{\text{tgt}}$ denotes the target latent representation (formerly referred to as the "original image").
> * $C_{\text{spg}}$ and $Z_{\text{idpg}}$ are now mathematically defined in **the newly added Equations (2) and (3)** to clearly show how conditions are concatenated.
>
> 3). **Structural Optimization:** We have divided the section into two distinct paragraphs, **"Seed Person Image Generator"** and **"Identity Preservation Generator"**, to allow for a more detailed and rigorously defined explanation of the input modifications for each module.
>
> **2. Novelty**
>
> We thank the reviewer for raising this crucial point regarding the novelty and positioning of the SPG. We clarify the distinct contributions of our approach compared to existing works as follows:
>
> 1). **VS. Aspect Ratio Bucketing (Standard Approach):**
> * **Existing Methods:** State-of-the-art models like SDXL [1], Flux [2] rely on "Aspect Ratio Bucketing", which groups training data into various resolution buckets. This paradigm necessitates massive datasets to ensure sufficient samples for each resolution bucket.
> * **Our Innovation (Data Efficiency):** In TPR, particularly under few-shot settings, such massive data is typically unavailable. Our SPG innovatively trains on **fixed-resolution** images while injecting original size parameters $(w_r, h_r)$ as the condition. This new method decouples the training resolution from the learned distribution, allowing the model to learn resolution *concepts* from limited data rather than merely memorizing discrete resolution *buckets*.
>
> 2). **Continuous Control & Grid Independence:**
> * **Existing Constraints:** Mainstream DiT-based [3] models (e.g., Flux [2]) enforce strict grid alignments (e.g., inputs must be multiples of 32 or 16 pixels) during both training and inference, limiting the flexibility of generating images with arbitrary resolutions or aspect ratios.
> * **Our Innovation (Infinite Control):** By embedding resolution as a semantic condition, our SPG enables **continuous (infinite) control** of image clarity and aspect ratio during inference. Notably, we can generate pedestrian images that simulate specific low-resolution surveillance scenarios or arbitrary aspect ratios, free from the constraints of the underlying DiT grid requirements.
>
> While the original manuscript includes a preliminary discussion in Section 3.1 (Line 161), we have further revised this paragraph following your suggestion to sharpen the contrast and highlight our novelty. Specifically, we have expanded the discussion on related generative methods and explicitly emphasized the unique advantages of our approach in data-scarce TPR scenarios.
>
> [1] Podell D, English Z, Lacey K, et al. Sdxl: Improving latent diffusion models for high-resolution image synthesis[J]. arXiv preprint arXiv:2307.01952, 2023.
>
> [2] BLACK FOREST LABS. Flux: Official inference repository for flux.1 models[CP/OL]. 2025[2025-11-26]. https://github.com/black-forest-labs/flux.
>
> [3] Peebles W, Xie S. Scalable diffusion models with transformers[C]//Proceedings of the IEEE/CVF international conference on computer vision. 2023: 4195-4205.

---

> ### Author Response · Authors · 2025-11-26
>
> We sincerely appreciate the reviewer’s thoughtful comment regarding the definition of the zero-shot setting. We fully agree that standard TPR protocols always involve non-overlapping identities between training and testing sets.
>
> However, we respectfully clarify that our definition of "zero-shot" aligns with the broader consensus in the literature on Transfer Learning and Vision-Language (e.g., CLIP [4]), rather than being limited to the specific "new identity" generalization within a single dataset.
>
> 1). **Definition Alignment:** In seminal works like CLIP [4], "zero-shot" refers to a model trained on large-scale external data (e.g., WebImageText) and evaluated directly on downstream tasks (e.g., ImageNet) **without seeing any training samples (images or labels) from the target dataset**.
>
> 2). **Our Setting:** Similarly, our method utilizes only a small set of real images to synthesize a completely new dataset. We then train the retrieval model exclusively on this synthetic data and evaluate it directly on the test sets of three benchmark datasets (CUHK-PEDES, ICFG-PEDES, RSTPReid). Since the model **never accesses the training splits** of these target datasets, this constitutes a zero-shot setting at the **dataset/domain level**, rather than only at the identity level.
>
> 3). **Precedents in Retrieval:** This definition is also standard in related retrieval tasks. For instance, in Zero-shot Composed Image/Video Retrieval [5, 6], approaches that synthesize training data and evaluate on target benchmarks without fine-tuning on the target training splits are strictly classified as zero-shot.
>
> To prevent any potential confusion for readers and to strictly distinguish our setting from standard supervised TPR (where models are trained on the target dataset's training split), we have added an explicit definition of our "zero-shot setting" in **Section 4.1 (Lines 332-333)** of the revised manuscript.
>
> All modifications have been marked in blue in the revised version. If you have any further questions, please do not hesitate to let us know.
>
> [4]Radford A, Kim J W, Hallacy C, et al. Learning transferable visual models from natural language supervision[C]//International conference on machine learning. PmLR, 2021: 8748-8763.
>
> [5]Ventura L, Yang A, Schmid C, et al. Covr: Learning composed video retrieval from web video captions[C]//Proceedings of the AAAI Conference on Artificial Intelligence. 2024, 38(6): 5270-5279.
>
> [6] Ventura L, Yang A, Schmid C, et al. CoVR-2: Automatic data construction for composed video retrieval[J]. IEEE Transactions on Pattern Analysis and Machine Intelligence, 2024.

---

### Meta-Review · Area_Chair_puRG · 2026-01-05

**Summary:**

The paper proposes TIPS, a framework for synthesizing text-image pairs to address data scarcity in Text-based Person Retrieval (TPR), particularly for zero-shot and few-shot scenarios. The framework utilizes two diffusion-based generators (Seed Person Image Generator and Identity Preservation Generator) and an MLLM-based filtering pipeline.

**Reviewer Concerns:**

The reviewers acknowledged the practical value of the problem and the comprehensiveness of the framework, with Reviewer 2BnY strongly endorsing the work. However, initial concerns were raised regarding the novelty (viewed by Reviewer Ao47 as engineering integration), methodological clarity (specifically the "zero-shot" definition and reliance on real images for training generators), computational cost, and the lack of quantitative validation for the generated data's quality (identity consistency and diversity).

The authors provided a robust rebuttal, adding significant new data: quantitative ReID metrics to prove identity consistency, background/pose diversity statistics, and extensive ablations on resolution control and MLLM filtering. However, Reviewer Ao47 remained critical regarding the paper's positioning (specifically the claim of reducing reliance on real data while still using 300 real images).

**Reviewer Scores:**

Reviewer Ao47 actively participated in the discussion but remained unconvinced by the authors' justifications regarding the zero-shot definition and the "no real data" claims, viewing them as logically inconsistent.

---

### Decision · Program_Chairs · 2026-01-26

Reject